# Anisotropic charge trapping in phototransistors unlocks ultrasensitive polarimetry for bionic navigation

Jing Pan[1,4], Yiming Wu[2,4], Xiujuan Zhang [1] ✉, Jinhui Chen[1], Jinwen Wang[1], Shuiling Cheng[1], Xiaofeng Wu[1], Xiaohong Zhang [1] ✉ & Jiansheng Jie [1,3] ✉

Being able to probe the polarization states of light is crucial for applications from medical diagnostics and intelligent recognition to information encryption and bio-inspired navigation. Current state-of-the-art polarimeters based on anisotropic semiconductors enable direct linear dichroism photodetection without the need for bulky and complex external optics. However, their polarization sensitivity is restricted by the inherent optical anisotropy, leading to low dichroic ratios of typically smaller than ten. Here, we unveil an effective and general strategy to achieve more than 2,000-fold enhanced polarization sensitivity by exploiting an anisotropic charge trapping effect in organic phototransistors. The polarization-dependent trapping of photogenerated charge carriers provides an anisotropic photo-induced gate bias for current amplification, which has resulted in a record-high dichroic ratio of $>10^4$, reaching over the extinction ratios of commercial polarizers. These findings further enable the demonstration of an on-chip polarizer-free bionic celestial compass for skylight-based polarization navigation. Our results offer a fundamental design principle and an effective route for the development of next-generation highly polarization-sensitive optoelectronics.

Polarization-sensitive photodetection capable of extracting polarimetric information beyond light amplitude and frequency promises major advances in technologies including polarization imaging[1,2], biological analysis[3], environmental monitoring[4], and navigation sensors[5,6]. Over past years, various strategies have been developed to realize the sensitive detection of linearly polarized light. A conventional way is to assemble polarization-insensitive photodiodes or cameras with spatially separated polarizers[7,8]. Although successful integration and commercialization have been achieved, this strategy requires bulky and complicated optical systems that add to the manufacturing cost. Furthermore, polarizers with high extinction ratios are usually fabricated at the cost of optical transmittance and imaging sensitivity.

Alternatively, on-chip polarimetry has been emerging as a promising candidate for polarization-sensitive photodetection. Such devices can be realized by either well designed plasmonic nanoantennas[9,10] or intrinsically anisotropic photoactive materials[11–13]. In particular, photoactive semiconductors with inherent optical anisotropy have aroused intensive attention due to the simplified and cost-effective route that combines light detection with polarization sensing in a single component. Thanks to the high aspect ratios or asymmetric crystal structures, the optical anisotropy of one-dimensional (1D) nanowires (NWs)[14–16] and nanotubes[17,18], two-dimensional (2D) layered materials[19–22], and perovskites[23–25] has been widely explored. Despite the advances achieved, research on these materials in highly polarization-sensitive photodetection is still in its

[1]Institute of Functional Nano & Soft Materials (FUNSOM), Jiangsu Key Laboratory for Carbon-Based Functional Materials & Devices, Soochow University, Suzhou, Jiangsu 215123, P. R. China. [2]Institute of Materials Research and Engineering, Agency for Science, Technology and Research (A*STAR), Singapore 138634, Singapore. [3]Macao Institute of Materials Science and Engineering, Macau University of Science and Technology, Taipa, Macau SAR 999078, P. R. China. [4]These authors contributed equally: Jing Pan, Yiming Wu. ✉e-mail: xjzhang@suda.edu.cn; xiaohong_zhang@suda.edu.cn; jsjie@suda.edu.cn

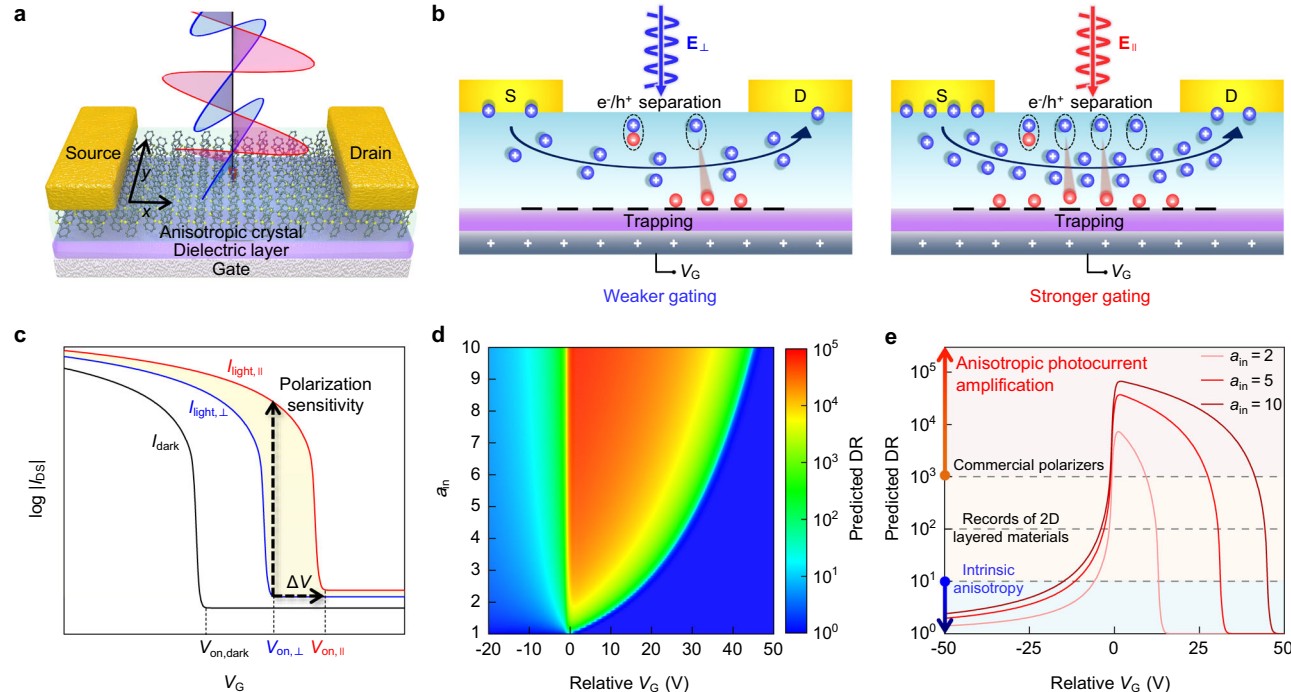

**Fig. 1 | Anisotropic charge trapping effect for polarization sensitivity enhancement in phototransistors. a** Schematic illustration of a bottom-gate top-contact phototransistor based on an anisotropic photoactive crystal. **b** Schematic illustrations of the anisotropic charge trapping effect. Anisotropic light absorption results in the polarization-dependent trapping of photogenerated charge carriers. These trapped carriers induce an anisotropic photo-induce gate bias to amplify the channel current to different extent. **c** Schematic illustration of the transfer characteristics of a p-type polarization-sensitive OPT. The shifts of onset/threshold voltage in different polarization states can lead to significant enhancement of polarization sensitivity. **d** Predicted relationship between $a_{in}$ and DR in an ideal OPT with a hole mobility of 1 cm$^2$ V$^{-1}$ s$^{-1}$. The relative $V_G$ represents the applied gate voltage relative to the reference threshold/onset point of $I_{light,\perp}$. **e** Predicted values of $a_{in}$-related DR enhanced by the anisotropic charge trapping effect.

infancy. For a photodetector that exhibits linear dichroism, the key figure-of-merit is the polarization sensitivity, which is usually represented by the dichroic ratio (DR) of maximum to minimum polarization-dependent photoresponse. However, in most cases, the obtained DRs were at a low level of <10 (ref. [15–17], [20–25]), which were fundamentally limited by the inherent anisotropy of photoactive materials and further weakened by the severe charge carrier recombination in photoconductors. In light of this, a few works have incorporated anisotropic materials into heterostructures to effectively separate the photogenerated charge carriers, thereby leading to enhanced DRs of ~10$^2$ (ref. [26], [27]). More recently, much improved polarization sensitivity has been further demonstrated by virtue of ferroelectrics or an external amplification circuitry[28,29]. Nonetheless, an effective and general strategy for anisotropic photocurrent amplification in single-component photodetectors remains elusive thus far, posing fundamental constraints to the promotion of simplified on-chip polarimetry.

In this study, we proposed a simple yet general anisotropic photocurrent amplification strategy to boost the polarization sensitivity of single-component phototransistors. Theoretical estimations unveiled the striking enhancement of DRs by several orders of magnitude owing to the anisotropic charge trapping effect. As a proof-of-concept, we demonstrated a highly polarization-sensitive organic phototransistor (OPT) based on 2,7-dioctyl[1]benzothieno[3,2-b][1]benzothiophene (C8-BTBT) crystal array with inherent optical anisotropy. By tuning the anisotropic trapping process of photogenerated charge carriers, a polarization-dependent photo-induced gate bias was generated to modulate the anisotropic photocurrent amplification. As a result, a >2000-fold enhanced DR of over 10$^4$ was achieved, comparable to the extinction ratios of commercial polarizers (>10$^3$). The combination of ultrahigh polarization sensitivity with the visible-blind ultraviolet (UV) response nature further enabled the OPT to mimic desert ants

for on-chip bionic navigation without external optics. Our work offers a feasible route towards high-performance on-chip polarimetric technologies.

## Results

### Prediction of the anisotropic charge trapping effect in phototransistors

The selective trapping of minority charge carriers is crucial to enhancing the photoconductivity of two-terminal photoconductors by prolonging the lifetime of majority charge carriers[30,31]. Specifically, the photogenerated charge carriers captured by the interfacial states can induce a localized electric field (photo-induced gate bias), thus effectively boosting the drain current ($I_{DS}$). This process can be further strengthened by harnessing the gate tunability of a three-terminal phototransistor, as the applied gate voltage ($V_G$) can adjust the number of trapped charge carriers and promote the current amplification in a more controllable manner[32]. On the other hand, in asymmetric crystal lattices, light absorption along different crystal axes can be anisotropic under polarized light[33,34]. A recent study has also unveiled the anisotropic charge transfer rate from organic crystals to polymer dielectrics under polarized illumination[35]. These facts imply that if abundant trap sites exist in a phototransistor composed of an anisotropic photoactive crystal (Fig. 1a), the number of trapped photogenerated charge carriers should also be polarization-dependent. Assuming a p-type photoactive crystal exhibits stronger polarized light absorption along its in-plane principal axis, light normally incident on the crystal with its e-vector (**E**) polarized parallel (||)/perpendicular (⊥) to the axis will result in the generation of more/fewer excitons (Fig. 1b). The more photogenerated excitons, the higher chance of electron trapping, thus the larger photo-induced gate bias. Consequently, the degree of current amplification can be highly dependent on light polarization states. We therefore speculate that the combination of charge trapping with

anisotropic light absorption might notably boost the polarization sensitivity of phototransistors. However, to the best our knowledge, the utilization of the anisotropic charge trapping effect for polarization sensitivity enhancement in phototransistors has yet to be explored. This may be largely due to the intricate charge trapping and detrapping processes, the insufficiently deterministic integration of semiconducting materials to charge trapping elements, and the unawareness of the potential significance of the anisotropic charge trapping phenomenon.

We first assessed the feasibility of the proposed photocurrent amplification strategy to achieve enormously enhanced polarization sensitivity from the device aspect. Figure 1c illustrates the typical transfer characteristics of a p-type OPT that exhibits the anisotropic charge trapping behavior. Due to the polarization-dependent trapping of photogenerated electrons, the onset/threshold voltages of the transfer curves drift towards the positive direction to different extent with varying the polarization states. Under a certain gate bias, while the drain current in the perpendicular state ($I_{light,\perp}$) is located at a subthreshold point, the drain current in the parallel state ($I_{light,\parallel}$) can rise several orders of magnitude above the threshold point due to a more positive shift of the onset/threshold voltage, thereby the DR ($I_{light,\parallel}$/$I_{light,\perp}$) is greatly enhanced.

We further performed theoretical simulations to evaluate the amplification of polarization sensitivity. The compact model of organic field-effect transistors (OFETs) and the photovoltaic mode of phototransistors were used to describe the drift of transfer curves under polarized light (see Supplementary Figs. 1–2 and Supplementary Tables 1–2 for more details). The incident light power ($P_{in}$)-dependent threshold voltage shift ($\Delta V_{th}$) in phototransistors can be expressed as[32,36,37]:

$$\Delta V_{th} = \frac{NkT}{q} \ln\left(1 + \frac{q\lambda\eta_{ex}P_{in}}{I_d hc}\right) = \frac{NkT}{q} \ln\left(1 + \frac{q\lambda\eta_{in}A_{abs}P_{in}}{I_d hc}\right) \quad (1)$$

where $N$ is an empirical parameter, $k$ is the Boltzmann constant, $T$ is the temperature, $q$ is the elementary charge, $I_d$ is the dark current of minority charge carriers, $\lambda$ is the wavelength of the incident light, $h$ is the Planck's constant, $c$ is the speed of light in vacuum, $\eta_{ex}$ and $\eta_{in}$ are the external and internal quantum efficiencies for photogeneration, respectively, and $A_{abs}$ is the optical absorbance that is polarization-dependent in anisotropic crystals[38,39]. In this expression, the product of $A_{abs}$ and $P_{in}$ represents the absorbed light power[40], which is influenced by either incident light power or light polarization state. For simplicity, we defined $a_{in}P_{eff}$ and $P_{eff}$ to equivalently simulate the absorbed light power in the parallel and perpendicular polarization states, respectively, where $P_{eff}$ is the effective light power and $a_{in}$ reflects the intrinsic anisotropic ratio of optical absorbance. Therefore, in an ideal case, the polarization-dependent threshold voltage shift ($\Delta V_{th,PD}$) can be predicted as:

$$\Delta V_{th,PD} = \Delta V_{th,\parallel} - \Delta V_{th,\perp} = \frac{NkT}{q}\left[\ln\left(1 + \frac{q\lambda a_{in}P_{eff}}{I_d hc}\right) - \ln\left(1 + \frac{q\lambda P_{eff}}{I_d hc}\right)\right] \quad (2)$$

where $\Delta V_{th,\parallel}$ and $\Delta V_{th,\perp}$ are the threshold voltages in the parallel and perpendicular polarization states, respectively. For instance, a photoactive crystal with $a_{in} = 5.6$ (which is the value of the organic crystal we used for fabricating the OPT in following discussions) may cause a $\Delta V_{th,PD}$ up to 34.2 V (Supplementary Fig. 3). Accordingly, a large difference in the trapped electron density of ~$3.8 \times 10^{12}$ cm$^{-2}$ was estimated on a 200 nm-thick SiO$_2$ dielectric layer ($\Delta N_{trap} = \Delta V_{th,PD}C_i/q$[41], where $C_i$ represents the unit-area capacitance of the dielectric layer), indicating the strong anisotropy of charge trapping under orthogonally polarized light. By substituting $a_{in}$-related $\Delta V_{th,PD}$ into the compact model of OFETs, we predicted the value of $I_{light,\parallel}$ relative to $I_{light,\perp}$, and

further obtained $a_{in}$-related DR (Fig. 1d). Compared with conventional polarization-sensitive photodetectors where DR is constrained by the intrinsic anisotropy, a tremendous improvement of DR by orders of magnitude can be achieved through anisotropic charge trapping in phototransistors (Supplementary Fig. 4a, the influence of mobility is discussed in Supplementary Fig. 4b). E.g., for commonly reported anisotropic crystals with $a_{in}$ of typically <10 (Supplementary Table 3), it is possible to get a strikingly high DR within a range of $10^3$–$10^5$, which far exceeds the records of 2D material-based photodetectors and approaches those of commercial polarizers (Fig. 1e).

## Growth and characterizations of the anisotropic organic crystals

To validate our hypothesis, we designed the OPT based on crystals of a small-molecule organic semiconductor C8-BTBT as a model device. C8-BTBT is an essential building block for high-performance organic electronics and optoelectronics due to its merits of high mobility, easy processing, good air stability, and large light absorption coefficient[42–45]. A solvent-free channel-restricted molecular flow method was developed to guide the ordered growth of C8-BTBT crystals in SiO$_2$ micro-channels (Supplementary Figs. 5–8). Angle-resolved cross-polarized optical microscopy (CPOM) and scanning electron microscopy (SEM) images unveil the formation of large-area highly aligned C8-BTBT crystal array (Fig. 2a, b). Atomic force microscopy (AFM) reveals the slight height difference (~9 nm) between the C8-BTBT crystal and the top surface of the SiO$_2$ channel wall (inset of Fig. 2b). The regular shape of the C8-BTBT crystal was further confirmed by the cross-sectional SEM image in Fig. 2c, which has a width of ~1 μm and a thickness of ~90–100 nm that is well filled in the SiO$_2$ micro-channel. The in-plane crystallographic information of C8-BTBT crystals was characterized by transmission electron microscopy (TEM) and high-resolution AFM (Fig. 2d, e). Both selected-area electron diffraction (SAED) patterns (inset of Fig. 2d and Supplementary Fig. 9) and fast Fourier transform (FFT) patterns (inset of Fig. 2e and Supplementary Fig. 10) verify the high crystal quality of C8-BTBT crystals with a herringbone-type molecular packing structure. The average values of lattice constants $a$ and $b$ obtained from 45 FFT patterns are $6.11 \pm 0.02$ Å and $8.26 \pm 0.07$ Å (Supplementary Fig. 11), respectively, which are well-consistent with previous report[46].

Thanks to the inherent herringbone-type molecular arrangement, C8-BTBT crystals should exhibit remarkable anisotropy[47]. Density functional theory calculations indicate that the transition dipole moment ($\mu$) from the highest occupied molecular orbital (HOMO) to the lowest unoccupied molecular orbital (LUMO) under excitation is nearly parallel to the π plane of the C8-BTBT molecule (Supplementary Fig. 12a and Supplementary Table 4). Therefore, the in-plane arrangements of $\mu$ (red arrows in Fig. 2f) are determined by the herringbone-stacked C8-BTBT molecules, resulting in anisotropic dipole strength along different crystal axes. Polarized absorption spectra further confirm the strong optical anisotropy of the C8-BTBT crystal (Fig. 2g). The obvious blue shift of peak positions (~0.04 eV) from parallel (0°) to perpendicular (90°) states is due to Davydov splitting (Supplementary Fig. 12b), which is characteristic of herringbone-type molecules[33]. Intriguingly, the highest absorption anisotropic ratio (~5.6, Supplementary Fig. 13) is located at 365 nm, which is a synergistic consequence of both the anisotropic peak position and intensity.

## Demonstration of ultrasensitive polarimetry in organic phototransistors

We fabricated a bottom-gate top-contact OPT based on the C8-BTBT crystal array (Supplementary Fig. 14) and assessed its performance under unpolarized light first. Notably, the OPT is highly photosensitive under 365 nm UV light with a high photosensitivity of over $10^6$, a high responsivity ($R$) of $1.7 \times 10^5$ A W$^{-1}$, a low noise equivalent power (NEP) down to 0.4 fW Hz$^{-0.5}$ at a modulation frequency of 1 Hz, and a high specific detectivity ($D^*$) of $1.1 \times 10^{13}$ Jones (Supplementary Figs. 15 and

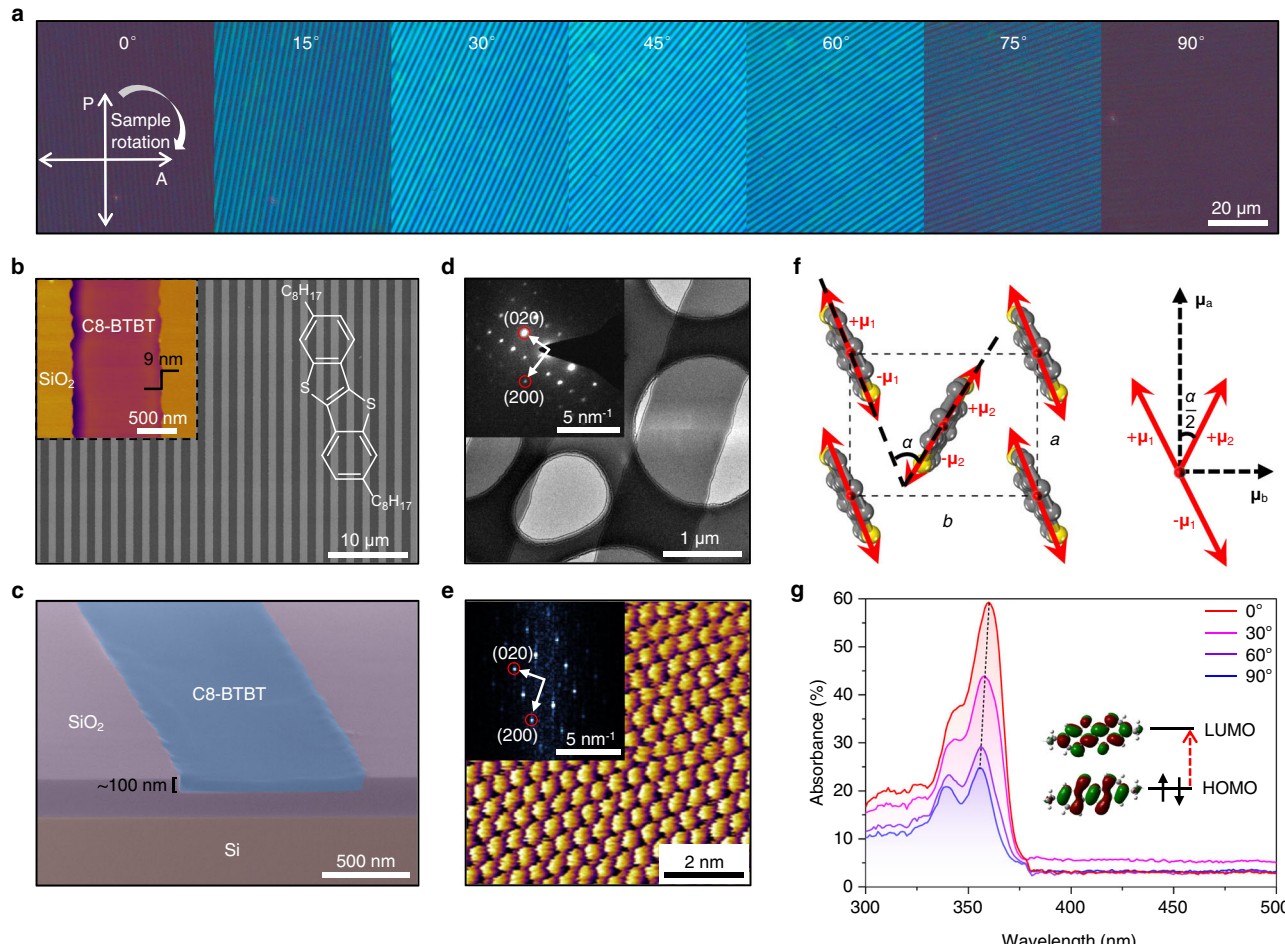

**Fig. 2 | Morphological characterizations and optical anisotropy of C8-BTBT crystals. a** Angle-resolved CPOM images of the C8-BTBT crystal array. **b** SEM image of the C8-BTBT crystal array. The darker ribbons indicate the C8-BTBT crystals while the brighter regions represent SiO₂ surface. The right inset shows the structure of a C8-BTBT molecule, and the left inset is an AFM image of a C8-BTBT crystal inside the SiO₂ channel. **c** Colored cross-sectional SEM image of a C8-BTBT crystal inside the SiO₂ channel in a 5° oblique view. **d** TEM image of a C8-BTBT crystal. The inset SAED pattern clearly reveals the diffraction spots of ($l$00) and ($0l$0) crystal planes. **e** High-resolution AFM scan of a C8-BTBT crystal. The inset shows the corresponding FFT pattern. **f** Schematic illustrations of the in-plane projections of **μ** (red arrows) in a C8-BTBT crystal lattice and the vector component of **μ** (black dashed arrows) along different crystal axes. **g** Polarized absorption spectra of a C8-BTBT crystal grown on quartz substrate. The inset is the calculated molecular orbitals of C8-BTBT, indicating the overlap of electron clouds at the molecular π cores. Electronic transitions from HOMO to LUMO (red arrow) can happen upon UV light absorption.

16a–c). The appealing photodetection performance of the OPT can be attributed to the photo-induced gating behavior[30], as reflected by the UV intensity-dependent $\Delta V_{th}$ (Supplementary Fig. 16d). Meanwhile, compared with the negligible hysteresis in the dark, the transfer curve of the OPT has an obvious memory window ($\Delta V_{th}$ = -19 V) under UV illumination with the change of gate sweeping direction (Supplementary Fig. 16e), indicating the trapping of photogenerated charge carriers upon illumination. We note that the filling of trap sites is associated with the rise of $I_{DS}$ when switching UV light on[48], which shows a unique photoadaptation characteristic that is tunable by both light intensity and the applied $V_G$ (Supplementary Figs. 16f and 17). Time-dependent decay behaviors further reveal the persistent photoconductivity caused by the gradual detrapping process (Supplementary Fig. 16f). It has been reported that the electrochemically active groups at the organic semiconductor/dielectric interface (e.g., –OH on SiO₂ surface) can act as electron trap sites[49,50]. Control experiments also reveal the crucial role of the C8-BTBT/SiO₂ interface in improving the photoresponse behaviors (Supplementary Fig. 18). The existence of interfacial trap sites on SiO₂ was further confirmed by scanning kelvin probe force microscopy (Supplementary Figs. 19 and 20). The scanning time-dependent potential drop on SiO₂ surface indicates the accumulation of negative charges. In contrast, the potential on trap-

free perfluoropolymer CYTOP remains unchanged with time. Notably, the active groups at C8-BTBT/SiO₂ interface have considerable stability and cannot be fully eliminated even in vacuum, as evidenced by the remarkable photoresponse of the OPT both in vacuum and air conditions (Supplementary Fig. 21). Apart from the –OH groups on SiO₂, the intentionally introduced trapping layer (e.g., nanoparticles/quantum dots[51] or donor/acceptor bulk heterojunctions[43]) can also be used for highly photosensitive OPTs.

Next, transfer characteristics of the OPT were measured under polarized light to unveil the feasibility of the anisotropic charge trapping effect. The light-triggered drain current ($I_{light}$) has an alternate change with the rotation of light polarization angle by every 90° (Fig. 3a). The highest $I_{light}$ is at polarization angles of 0°, 180°, and 360° while the lowest $I_{light}$ appears at 90°, 270°, and 450°, which refer to the polarization directions parallel and normal to the crystallographic $a$ axis of C8-BTBT, respectively. Significantly, an ultrahigh DR of $1.2 \times 10^4$ is obtained when $I_{light,\perp}$ is located at the threshold point with $I_{light,\parallel}$ rising steeply above the threshold, which still retains ~$8.6 \times 10^3$ even after preservation in ambient air for over 13 months (Supplementary Fig. 22). This tremendous enhancement of DR is in good agreement with the theoretically predicted results. As further evidence, $\Delta V_{th}$ changes periodically with light polarization angle (Fig. 3b). The

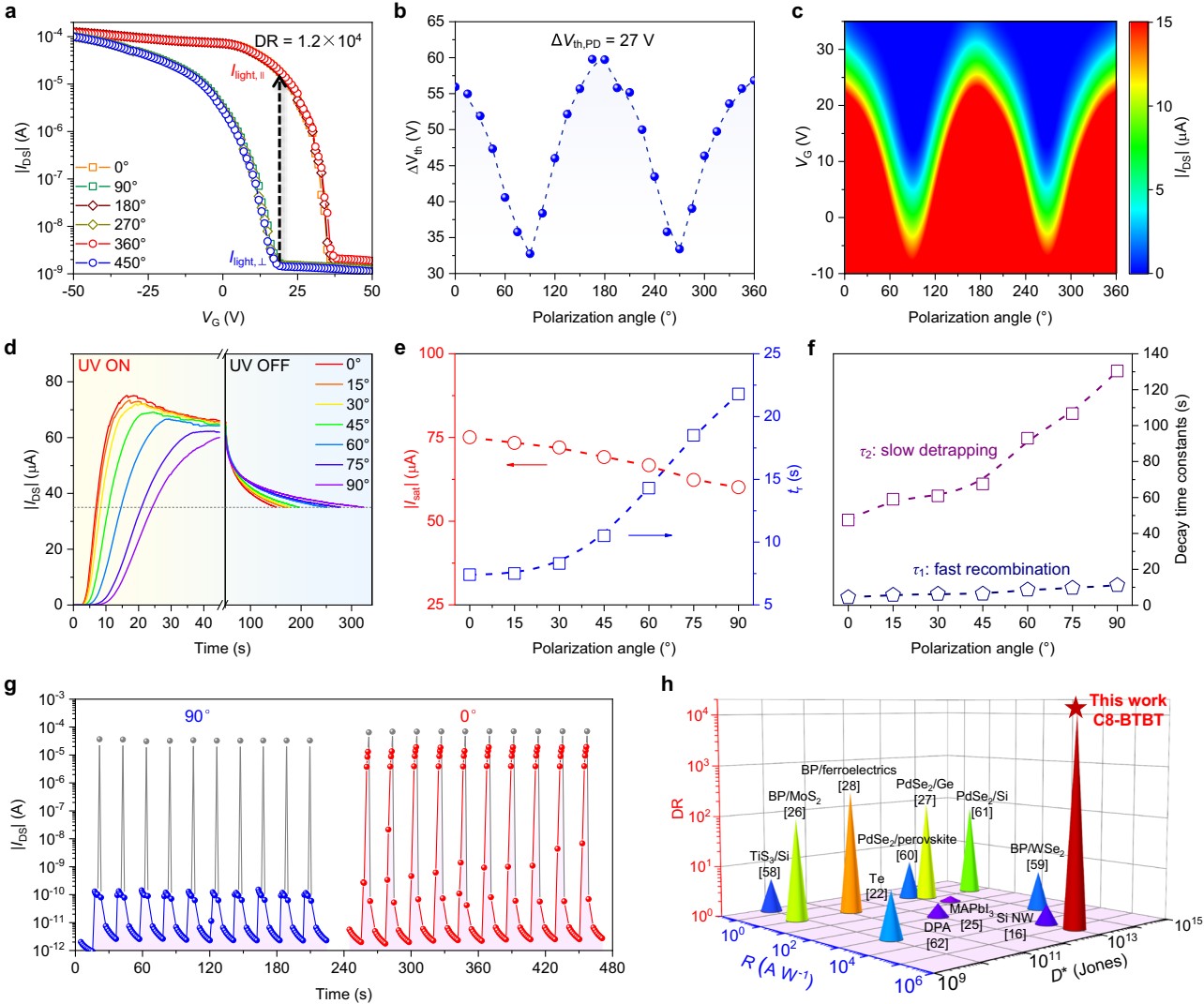

**Fig. 3 | Polarization-sensitive photodetection of the C8-BTBT crystal array.**
**a** Polarization-dependent transfer curves of the OPT under the illumination of
365 nm polarized light with a fixed intensity of 110 µW cm$^{-2}$ ($V_{DS} = -40$ V). **b** Periodic
variation of $\Delta V_{th}$ with the change of polarization angles. **c** Smoothed contour plot
of polarization-dependent $I_{DS}$ versus $V_G$ ($V_{DS} = -40$ V). **d** Time-related evolution of
$I_{DS}$ upon turning on/off the polarized light ($V_G = 25$ V, $V_{DS} = -40$ V). **e** Dependence of
$I_{sat}$ and $t_r$ on polarization angle. $t_r$ is defined as the time required for $I_{light}$ to increase

from 10%$I_{sat}$ to 90%$I_{sat}$. **f** Dependence of decay time constants $\tau_1$ and $\tau_2$ on polar-
ization angle. **g** Repeatable transient photoresponse of the OPT at polarization
angles of 90° (blue) and 0° (red). The gray data points refer to the response to an
erasing $V_G$ of −100 V in the dark. **h** Comparisons of DR, R, and D* values of the OPT
with other state-of-the-art polarization-sensitive photodetectors. More details are
available in Supplementary Tables 5 and 6.

difference between $\Delta V_{th,\parallel}$ and $\Delta V_{th,\perp}$ is ~27 V, which is close to the
predicted upper limit of $\Delta V_{th,PD}$ (-34.2 V, Supplementary Fig. 3b). We
note that the synergetic effect of charge trapping and anisotropic light
absorption is responsible for the enhanced polarization sensitivity. In
control experiments, although the OPTs based on C8-BTBT polycrystal
array with varied crystal orientations and amorphous C8-BTBT film
both exhibit remarkable photoresponse due to the pronounced photo-
induced gating behavior (Supplementary Fig. 23), the lack of aniso-
tropic light absorption leads to negligible polarization sensitivity.

Owing to the polarization-dependent threshold voltage, the
polarization sensitivity of the OPT can be readily tuned by applying an
appropriate gate voltage. Figure 3c depicts the contour plot of
polarization-dependent $I_{light}$ versus $V_G$ (polar plots of $I_{light}$ at different
$V_G$ are shown in Supplementary Fig. 24). The DR decreases with $V_G$
changing negatively (e.g., the DR is only ~2.6 when $V_G = -10$ V), because
when $V_G$ locates at the above-threshold, $I_{light}$ will maintain a high level
regardless of the polarization angle due to the injection and transport
of a large number of holes. In contrast, holes are gradually depleted
when $V_G$ becomes more positive, the influence of the photo-induced

gate bias is thus much more obvious. E.g., while $I_{light,\perp}$ is depleted in the
subthreshold region at $V_G = 20$ V, $I_{light,\parallel}$ can still be amplified to a high
level above the threshold due to the polarization-dependent $\Delta V_{th}$,
giving rise to a high DR of over $10^4$. In the meantime, we derived $V_G$-
dependent transconductance ($g_m$) to analyze the gate-controlled cur-
rent amplification (Supplementary Fig. 25a, b). The increase of $g_m$ with
$V_G$ becoming more negative is due to the reduced hole injection barrier
when the OPT is switched on. On the other hand, with the existence of
trap sites, $g_m$ becomes saturated and decreases at a higher negative $V_G$.
Intriguingly, while the initial dark-state $g_m$ has no anisotropy, the gate
voltages where $g_m$ starts to increase ($V_{st}$) and reaches its peak ($V_{sa}$)
exhibit a polarization-dependent trend similar to that of $\Delta V_{th}$ under
polarized light (Supplementary Fig. 25c, d). Furthermore, the peak $g_m$ is
also polarization-dependent with $I_{light,\parallel}$ rising more steeply than $I_{light,\perp}$
(Supplementary Fig. 25e), which is possibly caused by the increased
charge carrier concentration in the conductive channel and the
enhanced gating effect under stronger light absorption. Accordingly,
we reason that this unique anisotropic amplification characteristic
further endows the OPT with greatly boosted polarization sensitivity.

Apart from the gate voltage, the polarization sensitivity is also dominated by the number of the available trap sites (Supplementary Fig. 26). When a preset erasing gate voltage ($V_G$ = −60 V) is applied in the dark to detrap electrons for ensuring sufficient empty trap sites, the OPT exhibits higher polarization sensitivity (maximum DR = ~$10^4$). However, by programming the OPT at a positive $V_G$ of 25 V under illumination, the empty trap sites are partially filled by the photo-generated electrons. In this case, the OPT has much inferior polarization sensitivity (maximum DR = ~$10^2$). As explained in Supplementary Fig. 27, when electrons are detrapped and swept out by an erasing gate voltage, the OPT is in an unsaturated state where sufficient empty trap sites can trap the dissociated electrons upon illumination. Therefore, the number of newly trapped electrons as well as the photo-induced gate bias is highly polarization-dependent; weaker gating will result in fewer injected holes, while stronger gating will induce a larger number of holes and much stronger current amplification, leading to a larger DR. In contrast, when most of the trap sites are filled under illumination, the OPT is nearly in a saturated state where the enriched trapped electrons can initially induce a high channel current. Consequently, the strong gating effect remains in the OPT despite the change of polarization angle as the newly trapped photogenerated electrons account for only a small portion, thereby causing a smaller DR. It is noteworthy that the initial state of the OPT remains stable and repeatable with negligible $\Delta V_{th}$ after either erasing (Supplementary Fig. 28) or programming (Supplementary Fig. 29) in the dark, indicating that the polarization effect is only light-triggered, and the dark current is polarization-independent with no extra influence on the anisotropic charge trapping effect.

We further note that, due to the gradual filling of trap sites by photogenerated charge carriers, the polarization sensitivity of the OPT is illumination time-dependent. Figure 3d shows the time-related transient photoresponse of the OPT at different polarization angles. The rise time ($t_r$) is much shorter when the incident light is 0° polarized (~7.4 s) compared with the case where ~21.8 s is needed at 90° at a fixed UV intensity of 110 μW cm$^{-2}$ (blue curve in Fig. 3e), which makes a much faster increase of $I_{light,\parallel}$ than that of $I_{light,\perp}$ upon illumination. Meanwhile, with increasing the illumination time, the saturation current ($I_{sat}$) at different polarization angles approaches to similar values with small anisotropy due to the gradual filling of finite trap sites (red curve in Fig. 3e). Given the highly anisotropic photoresponse speed of the OPT, we reason that the maximum polarization sensitivity can be acquired by applying a proper illumination time (Supplementary Fig. 30). We also notice that the OPT maintained polarization-dependent persistent photoconductivity after turning off light (Fig. 3d). The decay of $I_{DS}$ can be described by a bi-exponential equation with time constants $\tau_1$ and $\tau_2$, respectively, representing the time for fast electron-hole recombination and slow electron detrapping[52] (Fig. 3f and Supplementary Fig. 31). The relatively shorter electron detrapping time at 0° (i.e., equivalently at a higher light intensity) is related to the filling of energetically distributed trap sites. Because longer-lived deeper traps will be preferentially occupied upon illumination, they account for a slower detrapping rate under weaker light; when these deeper traps have already been occupied under stronger light, filling of more shallower traps will take place, leading to a faster detrapping rate[48,51]. Figure 3g shows the transient on/off switching behaviors of the OPT under orthogonally polarized light. Notably, the OPT not only shows an enormously high DR of over $10^4$ after ~5 s of UV illumination, but also can be readily switched off to its initial dark current level within 1 s by applying an erasing $V_G$ of −100 V upon turning off light (Supplementary Fig. 32), and it retains stable and repeatable polarized photoresponse within 10 on/off switching circles, showing good durability and excellent fatigue resistance.

Intriguingly, the illumination time-dependent polarization sensitivity endows the OPT with an extraordinary characteristic of self-adaptation to light intensity (Supplementary Fig. 33). While a maximum DR can be achieved within seconds under the illumination of relatively strong light, the gradual charge-storage accumulation also ensures a persistent increase of DR under weak light. E.g., the DR under weak UV light of 0.68 μW cm$^{-2}$ can also reach its maximum of over $10^4$ by prolonging the illumination time to ~3 min (Supplementary Fig. 34). Unlike conventional polarization-sensitive photodetectors which possess limited and unchanged DR, this unique dynamic self-adaptation process promises the fine-tuned polarization sensitivity adequate for multi testing environments, and is especially useful for ensuring ultrasensitive polarimetry under weak light, which could be potentially applicable in a variety of fields such as deep-space exploration, artificial visual perception, and intelligent information processing[53–56].

To validate the universality of our approach, we fabricated OPTs based on blade-coated 2,7-didecylbenzothienobenzothiophene (C10-BTBT) and 2,8-difluoro-5, 11-bis(triethylsilylethynyl) anthradithiophene (dif-TES ADT) microribbons on SiO$_2$/Si substrates. Notably, the OPTs based on C10-BTBT and dif-TES ADT exhibit remarkable DRs of $3.7 \times 10^4$ and $2.0 \times 10^4$ under 365 nm and 525 nm polarized light, respectively (Supplementary Fig. 35a–f). In addition, our anisotropic photocurrent amplification strategy could also be applicable in fluorescent organic crystals. E.g., a high DR of $6.4 \times 10^3$ was achieved in the OPT based on a single-crystal 1,4-bis(4-methylstyryl)benzene (BSB-Me) flake under 365 nm polarized light (Supplementary Fig. 35g–i). More importantly, we further demonstrated the feasibility of the anisotropic charge trapping effect on a different gate dielectric. Specifically, we fabricated a low-operation voltage OPT based on blade-coated C8-BTBT microribbons on high-$\kappa$ Al$_2$O$_3$ (Supplementary Fig. 36a–g). Thanks to the considerable charge trapping capability of Al$_2$O$_3$[49,57], the OPT has a high DR of $2.4 \times 10^3$ with a small $\Delta V_{th,PD}$ of ~2.1 V (Supplementary Fig. 36h, i). Given the high capacitance of the Al$_2$O$_3$ layer ($C_i = 1.1 \times 10^{-7}$ F cm$^{-2}$, Supplementary Fig. 36f), the polarization-dependent difference in the trapped electron density was estimated to be ~$1.4 \times 10^{12}$ cm$^{-2}$ on Al$_2$O$_3$, which is comparable to that on SiO$_2$. Based on the aforementioned experiments, we reason that a significantly boosted DR and much lower power consumption should be envisaged through further optimization of material choice, device fabrication, and wavelength selection in a series of follow-up works.

To benchmark the figure-of-merit parameters of our OPTs, we compare them with those of the representative polarization-sensitive photodetectors in literature[16,22,25–28,58–62] (Fig. 3h, more details are available in Supplementary Tables 5 and 6). Significantly, by harnessing the anisotropic photocurrent amplification strategy in phototransistors, we are able to overcome the constraints of the intrinsic anisotropy of photoactive materials. As central figure-of-merits representing polarization sensitivity, our enhanced DRs ($10^3$–$10^4$) outperform the current records ($10^2$, ref. 26–29) of all those photoactive material-based photodetectors, and even reaches over the extinction ratios (>$10^3$) of commercial polarizers, which is crucial for practical applications, particularly for the environments with partially polarized light (Supplementary Fig. 37). Furthermore, thanks to the highly photoresponsive nature of C8-BTBT and the significant photocurrent modulation capability of phototransistor, both $R$ and $D^*$ values of our device are advantageous over those of conventional polarization-sensitive photodiodes or photoconductors, thus greatly enhancing the detectable signals especially in low-light conditions.

## Celestial compass for bionic polarization navigation
By leveraging the ultrasensitive polarized UV response of the OPT based on C8-BTBT crystal array, we demonstrated its application in bionic celestial compass to mimic the polarization navigation of desert ant, *Cataglyphis* (Fig. 4a), which is able to sense weak skylight polarization by taking advantage of two sets of orthogonally aligned UV photoreceptors in its ommatidia[63,64] (i.e., horizontal regions A-D and perpendicular regions E-F in Fig. 4b). In contrast to conventional bio-inspired polarization navigation sensors that consist of bulky

and spatially separated UV light filter, polarizer, and polarization-insensitive photodiode[7] (Fig. 4c), the combination of the visible-blind UV response nature of C8-BTBT with the ultrahigh polarization sensitivity of the OPT enables a filterless, polarizer-free, and miniaturized route towards polarization navigation (Fig. 4d). The proposed biomimetic strategy for polarization navigation is based on the robust polarization mode in the sky (Fig. 4e, f), where the polarization of scattered sunlight is symmetrical about the solar meridian (SM). In particular, the polarization direction of **E** at zenith is perpendicular to the SM, and this rule stably exists despite the change of solar azimuth angle ($\Phi_s$)[8]. Therefore, one can attain the relative angle between the heading direction and SM ($\Phi_o$) by measuring the direction of **E** at zenith, and further track the direction of travel due to a regular change of SM with respect to the north direction.

To effectively detect the weak partially polarized skylight with a degree of linear polarization of typically <60% (i.e., anisotropic ratio is <4)[65], we encapsulated the OPT onto indium tin oxide (ITO)/glass substrate using a quartz coverslip to avoid the possible interference caused by the fluctuation of environmental humidity (Fig. 4g and Supplementary Fig. 38). Angle-resolved $I_{DS}$ was recorded stepwise by rotating the device clockwise with the 0° reference direction heading towards north. It is noteworthy that the OPT shows excellent anti-interference ability with robust polarization sensitivity in both clear (Fig. 4h–j) and cloudy (Fig. 4k–m) conditions. Furthermore, although the skylight intensity weakens with time due to the westward movement of the sun, the polarization signals still remain distinguishable. By setting base lines to eliminate the influence of light intensity change, we fitted the normalized angle-resolved $I_{DS}$ to determine the strongest photoresponse direction for navigating **E** and SM (Fig. 4j, m and Supplementary Fig. 39). The navigated directions of **E** are ~−14° and ~−16° relative to the 0° reference (compass-pointed north), respectively, which are well-consistent with the real-time theoretical values (**E** is respectively in a range of −18° to −13° and −21° to −15° relative to the north direction according to the real-time $\Phi_s$ in Suzhou, Supplementary Fig. 40). We further conducted a set of polarization navigation measurements at different times and in different weather conditions. Encouragingly, despite the slight interferences caused by cloud movement and the prolonged measuring time, the navigated directions of **E** can truly reflect the skylight polarization modes in a real environment (Supplementary Fig. 41 and Supplementary Table 7). Although we have limited our discussion to division-of-time measurements where the data was acquired sequentially in time, we have innovatively made a tentative exploration towards the on-chip fabrication of a filterless and polarizer-free polarization navigation sensor. In future, with the improved integration level, it is promising that information containing all polarization angles can be acquired simultaneously by a set of monolithically integrated device arrays. Benefiting from the ultrahigh polarization sensitivity and the excellent anti-disturbance, our miniaturized polarimeters could become a powerful auxiliary for autonomous navigation especially when geomagnetic or satellite signals are interfered.

## Discussion

In contrast to conventional polarimetric measurements where researchers set their sights on deliberately designed artificial structures that require either bulky macroscopic systems or complicated subwavelength nanostructures. Our work provides a simple yet versatile approach for the realization of ultrahigh polarization sensitivity (DR > $10^4$) in inherently anisotropic organic crystals, which greatly enhances their practical values for simplified on-chip polarimetric technologies[66]. It is worth noting that the tremendous amplification of polarization sensitivity in OPTs is achieved at the cost of photoresponse speed. However, the unique photoadaptive characteristic shows its inherent advantages over conventional photodiodes in mimicking the bionic visual perception process, which shares a similar self-adaptation speed from several seconds to tens of seconds to light intensity. Particularly, the prominent photocurrent amplification capability makes the OPT especially applicable in artificial cognitive systems for acquiring weak signals in low-light environments[67]. Looking forward, we anticipate that this general strategy may also be implemented in the ultrasensitive detection of circularly polarized light in chiral organic semiconductors[68], and further be extended to anisotropic inorganic semiconductors or organic/inorganic hybrid systems, as such future prospects for ultra-compact polarimetry are brightening.

In conclusion, our combined theoretical and experimental work has proven that the anisotropic charge trapping effect in phototransistors can be leveraged to overcome the fundamental constraints of intrinsic anisotropy in polarization-sensitive photodetection. By modulating the anisotropic charge trapping process in organic crystals, polarization sensitivity amplification of over 2,000 folds can be achieved, thereby leading to an unprecedentedly high DR up to $10^4$, which is over 2 orders of magnitude higher than the previously reported highest values of 2D material-based photodiodes, and reaches over the extinction ratios of commercial polarizers. The excellent and stable polarization sensitivity with visible-blind UV response nature make the OPT promising building block for implementation of bionic navigation under partially polarized skylight. Intriguingly, our anisotropic photocurrent amplification strategy is applicable in other kinds of photoactive crystals. In addition, the easy-processing of organic semiconductors offers an inherently inexpensive and flexible platform that is particularly suitable for scale-up applications. These findings not only lay the foundation for the design of next-generation ultrasensitive polarimeters, but also provide fresh perspectives for the realization of highly integrated optoelectronic systems for bionic polarization navigation, low-light cognition, artificial intelligent information processing, and potentially many others.

## Methods

### Preparation of SiO₂ micro-channels

Firstly, poly(methyl methacrylate) (PMMA, MicroChem A2) resist was spin-coated on a $1.5 \times 1.5$ cm² highly n-doped SiO₂/Si substrate (resistivity <0.02 Ω cm, with a 300 nm thermally grown SiO₂ layer). After baking the PMMA layer at 120 °C for 5 min, it was exposed by electron-beam lithography (EBL, Hitachi SU5000 equipped with the Raith 150 electron beam writer). During EBL, the traces of electron-beam followed the predesigned patterns in a computer program. After exposure, the whole substrate was immersed in a mixed solution of methyl isobutyl ketone (MIBK, Aladdin) and isopropanol (IPA, Aladdin) with a volume ratio of 1:3 for pattern developing. Reactive ion etching (RIE, Plasmalab 80 plus) was then used to etch the exposed part of SiO₂ with a depth of ~100 nm. Finally, the remaining PMMA layer was removed by acetone cleaning in a sonicator.

### Fabrication of C8-BTBT crystal array

A channel-restricted molecular flow method was developed for the liquid phase assembly of C8-BTBT. Lyophilic modification was first performed on the SiO₂/Si substrate by ozone treatment for 10 min to ease the flow of liquid. In control experiment, lyophobic modification was realized by baking the SiO₂/Si substrate with 2 μL trichloro(1H,1H,2H,2H-perfluorooctyl)silane (FTS, Sigma-Aldrich) in a vacuum chamber at 90 °C for 10 min (pressure <50 mbar). Before crystal growth, C8-BTBT powder (purchased from Luminescence Technology Corp.) was placed near the defined SiO₂ micro-channels and sandwiched between a $2 \times 2$ cm² quartz coverslip and the SiO₂/Si substrate. The SiO₂/Si substrate was then fixed on a hot plate with one side slightly tilted up by a glass spacer (~1 mm) and another side attached on a hot plate by adhesive tape. Free-flowing C8-BTBT liquid was acquired on lyophilic SiO₂ by heating the substrate to 130 °C, which would fill the confined geometry of the SiO₂ micro-channels by

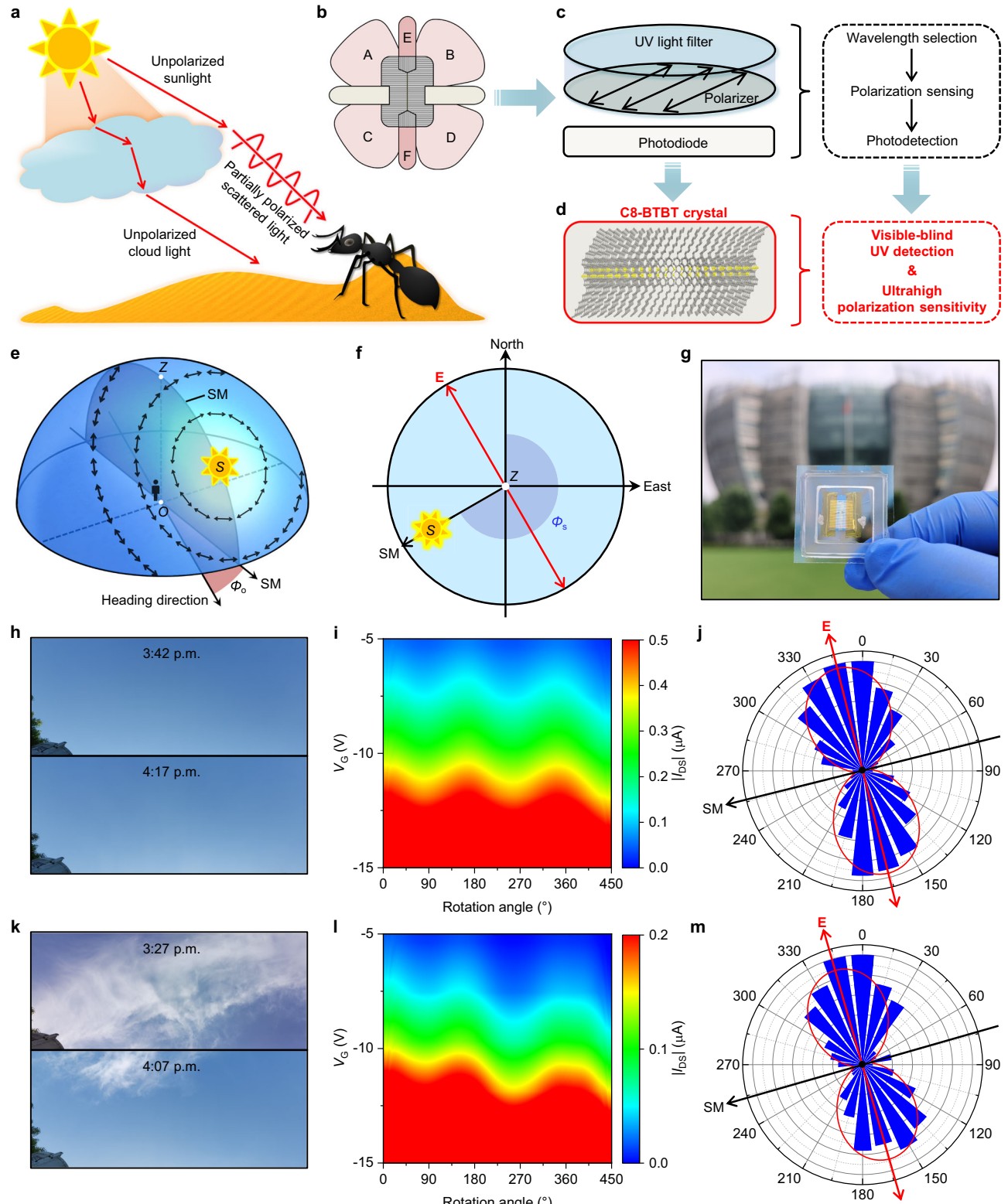

**Fig. 4 | Bionic polarization navigation implemented by C8-BTBT-based pho-totransistors. a** Schematic illustration of the polarization of skylight. **b** Cross-sectional sketch of the dorsal rim ommatidia of *Cataglyphis*. **c** Schematic illustration of a conventional polarization navigation sensor, which is composed of spatially separated light filter, polarizer, and photodiode. **d** Schematic illustration of the ordered molecular packing of C8-BTBT crystal, which combines visible-blind UV detection with ultrahigh polarization sensitivity in a single component. **e** Schematic illustration of the ideal skylight polarization mode with the direction of **E** marked by black arrows. The *O*, *Z*, and *S* points represent the positions of observer, zenith, and the sun, respectively. **f** 2D projection of the skylight polarization mode, where **E** at zenith is always perpendicular to the SM.
**g** Photograph of the C8-BTBT crystal array-based polarization navigation sensor after encapsulation. **h** Real-time photograph of the clear sky at 3:42–4:17 p.m. (23/09/2021). **i** Smoothed contour plot of angle-resolved $I_{DS}$ measured under clear sky. **j** Polar coordinate plot of normalized $I_{DS}$ measured under clear sky. **k** Real-time photograph of the cloudy sky at 3:27–4:07 p.m. (24/09/2021). **l** Smoothed contour plot of angle-resolved $I_{DS}$ measured under cloudy sky. **m** Polar coordinate plot of angle-resolved $I_{DS}$ measured under cloudy sky. The red and dark arrows represent the deduced directions of **E** and SM, respectively.

capillary flow to form well aligned crystals; while pinning of C8-BTBT liquid was realized on lyophobic SiO₂ surface to form polycrystals with varied crystal orientations in SiO₂ micro-channels. After capillary flow, the quartz was uncovered immediately to avoid sticking with the underneath SiO₂/Si substrate, and the whole substrate was then naturally cooled to room temperature for the crystallization of C8-BTBT.

## Morphological characterizations and optical anisotropy analyses

Morphological characterizations of C8-BTBT crystals were performed by using CPOM (Leica DM4P), SEM (Hitachi SU5000), and AFM (Asylum Research Cypher S). The in-plane crystal structure of C8-BTBT was characterized by TEM (FEI Tecnai G2 F20) and high-resolution AFM (Asylum Research Cypher S). Polarized light absorption measurements were conducted by using UV-Vis spectrophotometer (PerkinElmer LAMBDA 1050+) equipped with an auto-rotatable Glan-Thompson polarizer. Density functional theory calculations were performed using Gaussian 16 with B3LYP/6-31 G(d, p) basis set. The crystallographic information file of C8-BTBT was available at http://www.chemspider.com.

## Device fabrication and characterizations

Source/drain Au electrodes (50 nm) of the OPT were thermally deposited through a shadow mask at a low deposition rate of ~0.1 Å s⁻¹. All electrical characterizations of the OPT were carried out at room temperature in a vacuum probe station connected with a semiconductor parameter analyzer (Keithley 4200-SCS) unless otherwise mentioned. A 365 nm UV lamp was placed right above the window of the chamber for photoelectrical measurements, and the light intensity was calibrated by a light power meter (Newport 843-R) connected with a standard silicon photodiode (Newport 918D-UV-OD3R). A rotatable UV linear polarizer (Thorlabs LPUV100-MP2, extinction ratio $10^3$–$10^5$ in a wavelength rage of 365–395 nm) was set between the chamber and the UV lamp for polarization-sensitive photoelectrical characterizations, and the intensity of polarized UV light was fixed to be 110 μW cm⁻² unless otherwise mentioned. For polarization navigation measurements, the OPT was fixed on an ITO/glass substrate with Si back gate stuck to ITO electrode by gallium indium eutectic. Silver wires (20 μm in diameter) were used to bond the source/drain electrodes of the OPT on ITO electrodes. A $2.5 \times 2.5$ cm² quartz coverslip was used to encapsulate the OPT. The encapsulated device was fixed into a dark chamber with three alligator clips extending out and connected with a semiconductor parameter analyzer (Keithley 4200-SCS) for outdoor electrical measurements.

## Data availability

The data that support the findings of this study are available from the corresponding authors upon request. Technical details of anisotropic charge trapping estimation are provided in the Supplementary Information. Source data are provided with this paper.

## Code availability

Density functional theory calculations were performed using the Gaussian 16 Simulation Package[69]. The codes that support theoretical simulations within this paper are available from the corresponding authors upon request.

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

## Acknowledgements

This work was supported by the National Natural Science Foundation of China (Grant No. 52225303 and 51973147 to J.J., 51821002 and 91833303 to Xiao.H.Z., and 52173178 to Xiu.J.Z.), Suzhou Science and Technology Plan Forward-Looking Project (Grant No. SYG202023 to Xiu.J.Z.), the Postdoctoral Research Foundation of China (Grant No. 2020M681705 to J.W.), Suzhou Key Laboratory of Functional Nano & Soft Materials, Collaborative Innovation Center of Suzhou Nano Science & Technology, the 111 Project, and Joint International Research Laboratory

of Carbon-Based Functional Materials and Devices. Y.W. acknowledges the funding support (Career Development Fund No. C222812011) from the Agency for Science, Technology, and Research (A*STAR). The authors thank R. Jia, C. Wang, and S. Chen for their technical assistance and helpful discussion.

## Author contributions

J.P., Xiu.J.Z. and J.J. conceived and designed the experiments. J.P. did theoretical estimations and sample preparation with assistance from J.C. and J.W. Device fabrication, characterization, and photoelectrical measurements were performed by J.P., J.C. and X.W. Density functional theory calculations were conducted by S.C. The manuscript was written by J.P., Y.W., Xiu.J.Z. and J.J. All the authors contributed to the scientific discussion. Xiu. J.Z., Xiao. H.Z. and J.J. supervised the project.

## Competing interests

The authors declare no competing interests.
