## [Peer Review File · Nature Communications]

Title: Anisotropic charge trapping in phototransistors unlocks ultrasensitive polarimetry for bionic navigationREVIEWER COMMENTS

Reviewer #1 (Remarks to the Author):

This manuscript reports a careful study on polarization-sensitive phototransistors made of C8-BTBT crystal array, based on which an on-chip polarizer-free bionic navigation device is fabricated. C8-BTBT has been extensively studied as a high-mobility model molecule and its high-quality oriented crystals along with high-performance OFETs/OPTs have been reported by many groups. It is not surprising to see anisotropic absorption of carefully prepared crystals and the successful translation of anisotropic properties (with the help of charge trapping effect on the semiconductor/dielectric interface) into detectable electric signals is easy to predict. From this point of view, the novelty of this manuscript is not outstanding. On the other hand, technically this work is very complete and it shows single-component simple-structured devices could realize unprecedented polarization sensitivity (DR up to 10^4), which may indeed become applicable in real world (if the concerns listed below can be addressed). Therefore in terms of practical applications, it has its strong points.

Two concerns are asked to be addressed:

1. 'General and fundamental design principle' is emphasized throughout the manuscript. Though 2 more molecules, C10-BTBT and dif-TES ADT, are also studied, while the experimental evidence is not strong enough. Firstly, I don't see much difference between C8-BTBT and C10-BTBT, therefore C10-BTBT may not be a good choice to prove the universality of the concept, at least not the best two. Secondly, there's obvious decrease (about one order of magnitude) in DR values of devices based on C10-BTBT and dif-TES ADT, though these values are still 'remarkable'. What's more notable is that the modulation on threshold voltage of dif-TES ADT devices seems to be much weaker than that of BTBT molecules (Figure S32). If no reasonable interpretation can be provided on these phenomena, my impression is that the crystal quality (maybe also the intrinsic mobility) of the molecules under the given device fabrication process and setting up plays a key role, which in turn compromise the general applicability of the introduced strategy.
2. If we are talking about the real applications, DR is obviously not the only main focus. In other words, Figure 3g only shows part of the story and is somewhat misleading. There are several important aspects of the devices in this manuscript that need to be objectively demonstrated. For example, i) the realization of ultrahigh DR is mostly ascribed to small OFF current down to nA, which may impose difficulty on signal detection for practical use; ii) the operational voltages are generally as high as several tens of volts and the response time of the devices is quite slow; iii) though the nice picture of compact on-chip device with encapsulation is shown, it should still be carefully fixed and connected with a semiconductor parameter analyzer for outdoor measurements, which, from my point of view, is not more advanced/convenient compared with the examples described in the manuscript as 'requires bulky and complicated optical systems that add to the manufacturing cost', even on the contrary, more operatively complex.

Reviewer #2 (Remarks to the Author):

The paper reports on the polarization-sensitive phototransistor by applying an anisotropic charge trapping effect, realizing a high dichroic ratio. The key point to realize it is the interface and well-aligned organic single-crystal array. Moreover, the potential application for bionic navigation is demonstrated. The work is well conducted and clearly written. I would therefore suggest the publication of the work, after addressing the following questions.

1. Based on equation 1, ΔV_{th} is proportional to η_{AP} , while ΔV_{th} is also proportional to ΔN_{trap} . Does it mean all the photogenerated carriers are trapped? Is a conversion coefficient needed between the photogenerated carriers and trapped carriers?
2. In page 9, can the authors explain the reason that the trap site filling time is relatively long (>5s) and different from the light intensity?
3. Also, one can detrapp the electrons by applying an erasing gate voltage to achieve a fast recovery. How to increase the response speed or the trap speed?

Reviewer #3 (Remarks to the Author):

This paper uses the defect structure of the silicon dioxide layer on the silicon substrate to construct a facile charge trapping effect, which can be combined with the single crystal active layer to achieve the anisotropy charge trapping, which is very innovative and interesting, but there are still some problems in the mechanism analysis and experimental methods, and it is suggested to be published after major revision.

- 1, please confirm whether formula 1 is written correctly. More often, we use this formula to describe photocurrent (I_{ds}), which will be different with by one transconductance factor from the current expression.
2. The current formulae 1,2 are mainly concerned with the anisotropy of crystals. Why can we see the difference in anisotropy charge trapping from these formulas? Please explain further, for example, which parameter it belongs to.
3. In fact, the change of threshold voltage is relatively small, and the corresponding photocurrent (I_{ds}) change is very large, dose this result indicate that the transconductance effect of anisotropy is also different.
4. The completely different charge trapping characteristics of materials on SiO₂ and CYTOP are good evidence for the effect of interface defects. Is there further quantitative results, such as the difference of subthreshold swing amplitude and channel defect density between the two anisotropy charge trapping?

5. Why the charge capture using SiO₂ can only be seen under light? SiO₂ in dark state has defects. Whether the SiO₂ used in this paper has been processed or prepared by a special method.

6. Slow response is a big problem for this kind of device. The paper only gives rise time, what is the descent time of this kind of device?

Responses to the reviewers' comments (Manuscript number NCOMMS-22-17386)

Changes in the revised manuscript as a response to the reviewers' comments are highlighted in **red color** and clarifications regarding the reviewer's comments are provided in **blue color**.

Response to reviewer #1:

Reviewer's comments:

This manuscript reports a careful study on polarization-sensitive phototransistors made of C8-BTBT crystal array, based on which an on-chip polarizer-free bionic navigation device is fabricated. C8-BTBT has been extensively studied as a high-mobility model molecule and its high-quality oriented crystals along with high-performance OFETs/OPTs have been reported by many groups. It is not surprising to see anisotropic absorption of carefully prepared crystals and the successful translation of anisotropic properties (with the help of charge trapping effect on the semiconductor/dielectric interface) into detectable electric signals is easy to predict. From this point of view, the novelty of this manuscript is not outstanding. On the other hand, technically this work is very complete and it shows single-component simple-structured devices could realize unprecedented polarization sensitivity (DR up to $10E4$), which may indeed become applicable in real world (if the concerns listed below can be addressed). Therefore in terms of practical applications, it has its strong points.

Reply: We greatly appreciate the reviewer's positive comments on our work. Indeed, C8-BTBT is an ideal organic semiconductor having advantages of easy processing, good air stability, and excellent electrical and optoelectronic properties for high-performance devices. Additionally, the visible light-blind nature and excellent anisotropy in the UV spectral range make C8-BTBT particularly useful for bionic polarization navigation. This is why we choose C8-BTBT as a model photoactive material to construct our polarization-sensitive OPT.

In terms of the reviewer's concern on the strength of novelty of our work, we are sorry that we have not clarified the significance of our work clearly enough in our manuscript. Therefore, we would like to seize the opportunity to more clearly highlight and explain the specific challenges in the field of polarization-sensitive photodetection, and the reasons why our strategy is of great importance to the community and could advance the field and address some long-held limitations.

Thus far, the mainstream researches in the community have focused on the rational selection of highly anisotropic materials for seeking higher polarization sensitivity. Many state-of-the-art material systems, including 1D nanostructures, 2D layered materials, perovskites, and organic crystals have been systematically studied and carefully chosen for polarization-sensitive photodetectors. Nonetheless, limited by the inherent anisotropy of these materials, the obtained dichroic ratio (DR) is still at a low level of typically smaller than 10, which is insufficient for practical applications. Therefore, the effective enhancement of DR remains a central challenge for on-chip polarization-sensitive photodetectors, which has recently been explored by virtue of hybrid device structures such

as graphene/PdSe₂/Ge heterojunction (DR = 112, *ACS Nano* 2019, 13, 9907-9917), black phosphorus p-n homojunction with ferroelectric domains (DR = 288, *Nat. Commun.* 2022, 13, 3198), and an integrated amplification system combining a nanowire photodetector with an OFET (DR = 375, *Nat. Commun.* 2021, 12, 6476). However, there still lacks an effective and general strategy for polarization sensitivity amplification in single-component photodetectors, posing a fundamental challenge for the promotion of simplified on-chip polarimetry.

To address the challenge described above, we have shown that overcoming the inherent restriction of linear dichroism to achieve significantly boosted polarization sensitivity is possible through the exploitation of an anisotropic charge trapping effect in phototransistors. To the best of our knowledge, the utilization of the anisotropic charge trapping effect for polarization sensitivity enhancement in phototransistors has yet to be explored. Thus far, only a small number of studies have reported phototransistor-type polarization-sensitive photodetectors with limited DRs (e.g., DR = 2 in *Adv. Mater.* 2018, 30, 1804541; DR = 1.7 in *Small* 2021, 17, 2008078; DR = 1.8 in *Mater. Horiz.* 2022, 9, 1448-1459), leaving a huge gap between theory and experiment. This may be largely due to the lack of full consideration of the intricate charge trapping and detrapping processes, as well as the insufficiently deterministic integration of semiconducting materials to charge trapping elements. Since in conventional ways of thinking, the introduction of trap sites could be a negative factor that degrades the device performance, no previous works have been aware of the potential significance of the anisotropic charge trapping phenomenon.

In this work, for the first time, we systematically unveiled that the anisotropic trapping of photogenerated charge carriers in phototransistors can induce an additional polarization-dependent photo-induced gate bias to modulate the anisotropic photocurrent amplification. Our established theory unites the advantages of light sensitivity enhancement in phototransistors and anisotropic light absorption of photoactive crystals, and was successfully used to predict the dramatic polarization sensitivity enhancement. Using this strategy, we achieved a more than 2,000-fold enhanced polarization sensitivity in organic phototransistors by cleverly exploiting the naturally existing charge trapping effect on the organic semiconductor/SiO₂ dielectric interface without further destruction of the crystal quality, and obtained an unprecedented high DR of over 10⁴ that is more than two orders of magnitude higher than the largest value ever reported in 2D material-based photodiodes (~10²). Up to now, no solution has been found to provide such a dramatic enhancement in the polarization sensitivity. Our findings also provide a robust, general, and scalable solution for developing ultrasensitive polarimetry with simple device structure, and could facilitate the development of next-generation highly polarization-sensitive optoelectronics for bionic applications. To give a clearer explanation on the research background, we have further modified the corresponding sentence on Page 4, Lines 91-96 in the revised manuscript. We feel that these breakthroughs are of substantial importance to the photonics and electronics communities, and hope the reviewer concurs after going through our clarification marked in the revised manuscript.

Two concerns are asked to be addressed:

1. ‘General and fundamental design principle’ is emphasized throughout the manuscript. Though 2 more molecules, C10-BTBT and dif-TES ADT, are also studied, while the experimental evidence is not strong enough. Firstly, I don’t see much difference between C8-BTBT and C10-BTBT, therefore C10-BTBT may not be a good choice to prove the universality of the concept, at least not the best two.

Reply: We thank the reviewer for this valuable comment. To demonstrate the universality of our strategy, we have conducted a number of new experiments and fabricated the polarization-sensitive OPT based on 1,4-bis(4-methylstyryl)benzene (BSB-Me), a totally different fluorescent material without thiophene rings. Additionally, we acquired single-crystal BSB-Me flakes with regular shapes by drop casting of saturated solution, which also differ from those aligned crystal arrays and microribbons in morphology. The detailed results have been shown in Supplementary Fig. 35g-i in the revised manuscript. A remarkably high DR of 6.4×10^3 was achieved in the OPT based on a single-crystal BSB-Me flake. These results deterministically demonstrated that our anisotropic photocurrent amplification strategy could be applicable to different material systems to realize ultrasensitive polarimetry.

Supplementary Figure 35 | a, Molecular structure of C10-BTBT. b, CPOM image of C10-BTBT

microribbons on SiO₂/Si substrate. **c**, Polarization-dependent transfer curves of the OPT based on C10-BTBT microribbons under 365 nm polarized light with an intensity of 110 μW cm⁻² ($V_{DS} = -40$ V). **d**, Molecular structure of dif-TES ADT. **e**, CPOM image of dif-TES ADT microribbons on SiO₂/Si substrate. **f**, Polarization-dependent transfer curves of the OPT based on dif-TES ADT microribbons under 525 nm polarized light with an intensity of 11.6 μW cm⁻² ($V_{DS} = -40$ V). **g**, Molecular structure of BSB-Me. **h**, CPOM image of a BSB-Me flake on SiO₂/Si substrate. **i**, Polarization-dependent transfer curves of the OPT based on a BSB-Me flake under 365 nm polarized light with an intensity of 110 μW cm⁻² ($V_{DS} = -40$ V). In all these measurements, the maximum photoresponse direction is set as the 0° reference. Both C10-BTBT and dif-TES ADT microribbons were prepared by a blade-coating method²³, and the narrow-bandgap dif-TES ADT microribbons were decorated with 10 nm wide-bandgap C8-BTBT via thermal evaporation to block undesired electron injection in the dark¹⁵. The BSB-Me flake was acquired through drop casting the saturated solution on SiO₂/Si substrate.

Secondly, there's obvious decrease (about one order of magnitude) in DR values of devices based on C10-BTBT and dif-TES ADT, though these values are still 'remarkable'. What's more notable is that the modulation on threshold voltage of dif-TES ADT devices seems to be much weaker than that of BTBT molecules (Figure S32). If no reasonable interpretation can be provided on these phenomena, my impression is that the crystal quality (maybe also the intrinsic mobility) of the molecules under the given device fabrication process and setting up plays a key role, which in turn compromise the general applicability of the introduced strategy.

Reply: We thank the reviewer for raising this concern. In terms of lower DR values of the OPTs based on C10-BTBT and dif-TES ADT microribbons, we reason that the crystal growth and device fabrication processes were not optimized in the previous version of our manuscript (especially for the non-perfectly aligned dif-TES ADT microribbons), and these issues could indeed lead to degraded DRs. However, on the other hand, these results are still good evidence to support the robustness of our anisotropic photocurrent amplification strategy, showing that the lower limit of DR could still reach over 10³ even with casually fabricated OPTs. To address the potential concerns of the readership, we have modified our devices, including improving the alignment uniformity of the microribbons and scratching the device margins to eliminate the possible crosstalk by adjacent materials. As a result, the DR of the C10-BTBT-based OPT was improved to 3.7×10⁴ (see Supplementary Fig. 35a-c in the revised manuscript).

Regarding the weak modulation on threshold voltage of the dif-TES ADT-based OPT in our previous version, we reason that besides the alignment issue, the minority charge carrier injection effect plays a critical role (*Adv. Funct. Mater.* 2019, 1906653; *Nanoscale Horiz.* 2020, 5, 454-472). The relatively small bandgap of dif-TES ADT (~2.1 eV) would cause undesired electron injection in the dark and a much more positively drifted transfer curve (dark-state onset voltage = ~24 V). Since trap sites have already been partially filled in the dark, fewer electrons could be trapped under

illumination, leading to a smaller threshold voltage shift. Following our previous work (*Adv. Funct. Mater.* 2021, 31, 2100202), 10 nm C8-BTBT (bandgap = ~3.3 eV) was thermally evaporated on dif-TES ADT microribbons to block excess electron injection, thus improving the bias stress stability of dif-TES ADT-based OPT (dark-state onset voltage = ~6 V). Note that C8-BTBT only acts as an electron injection blocking material, and it does not influence the visible light response of dif-TES ADT as its absorption cutoff edge is located at ~370 nm. As shown in Supplementary Fig. 35d-f in the revised manuscript, with further improved alignment uniformity, the OPT based on C8-BTBT-decorated dif-TES ADT also has an ultrahigh DR of 2.0×10^4 .

Accordingly, we have modified the corresponding sentences on Pages 10-11, Lines 287-294 in the revised manuscript: “To validate the universality of our approach, we fabricated OPTs based on blade-coated 2,7-didecylbenzothienobenzothiophene (C10-BTBT) and 2,8-difluoro-5, 11-bis(triethylsilylethynyl) anthradithiophene (dif-TES ADT) microribbons on SiO₂/Si substrates. Notably, the OPTs based on C10-BTBT and dif-TES ADT exhibit remarkable DRs of 3.7×10^4 and 2.0×10^4 under 365 nm and 525 nm polarized light, respectively (Supplementary Fig. 35a-f). In addition, our anisotropic photocurrent amplification strategy could also be applicable in fluorescent organic crystals. *E.g.*, a high DR of 6.4×10^3 was achieved in the OPT based on a single-crystal 1,4-bis(4-methylstyryl)benzene (BSB-Me) flake under 365 nm polarized light (Supplementary Fig. 35g-i).”

Significantly, we further conducted experiments to prove that our strategy is also applicable to the OPT containing a high- κ dielectric layer, showing its compatibility with other substrates (see the following replies for details). In closing, we have added a corresponding summary on Page 11, Lines 301-304 in the revised manuscript: “Based on the aforementioned experiments, we reason that a significantly boosted DR and much lower power consumption should be envisaged through further optimization of material choice, device fabrication, and wavelength selection in a series of follow-up works.”

2. If we are talking about the real applications, DR is obviously not the only main focus. In other words, Figure 3g only shows part of the story and is somewhat misleading.

Reply: We thank the reviewer this valuable comment. We agree with the reviewer that other figure-of-merit parameters of our OPT should also be taken into consideration. Therefore, we have reorganized Fig. 3 and compared two central parameters of photodetectors (*i.e.*, photoresponsivity and detectivity), details are available in Fig. 3h and Supplementary Tables 5 and 6 in the revised manuscript.

Fig. 3 | Polarization-sensitive photodetection of the C8-BTBT crystal array. h, Comparisons of DR, R , and D^* values of our OPTs with other state-of-the-art polarization-sensitive photodetectors. More details are available in Supplementary Tables 5 and 6.

Meanwhile, we have modified the corresponding sentences on Page 11, Lines 305-316: “To benchmark the figure-of-merit parameters of our OPTs, we compare them with those of the representative polarization-sensitive photodetectors in literature^{16,22,25-28,58-62} (Fig. 3h, more details are available in Supplementary Tables 5 and 6). Significantly, by harnessing the anisotropic photocurrent amplification strategy in phototransistors, we are able to overcome the constraints of the intrinsic anisotropy of photoactive materials. As central figure-of-merits representing polarization sensitivity, our enhanced DRs (10^3 - 10^4) outperform the current records (10^2 , ref. ²⁶⁻²⁹) of all those photoactive material-based photodetectors, and even reaches over the extinction ratios ($> 10^3$) of commercial polarizers, which is crucial for practical applications, particularly for the environments with partially polarized light (Supplementary Fig. 37). Furthermore, thanks to the highly photoresponsive nature of C8-BTBT and the significant photocurrent modulation capability of phototransistor, both R and D^* values of our device are outstanding among those of conventional polarization-sensitive photodiodes or photoconductors, thus greatly enhancing the detectable signals especially in low-light conditions.”

There are several important aspects of the devices in this manuscript that need to be objectively demonstrated. For example, i) the realization of ultrahigh DR is mostly ascribed to small OFF current down to nA, which may impose difficulty on signal detection for practical use;

Reply: We thank the reviewer for raising this concern. We agree that one possible route to increase DR is to lower the off-state current. However, it should also be noted that one of the most important advantages of our OPT over photodiodes and photoconductors is its significant photocurrent amplification capability. Since the on-state current is greatly enhanced to tens of μ A under polarized light, we reason that it plays a crucial role in the enhancement of DR. Accordingly, we have searched in literature to provide statistical evidence that shows the possible currents acquired in the maximum

and minimum polarized photoresponse directions (denoted as $I_{\text{light},\parallel}$ and $I_{\text{light},\perp}$, respectively). As shown in Table R1, $I_{\text{light},\parallel}$ of our OPT (21,000 nA) outperforms those of the currently reported state-of-the-art polarization-sensitive photodetectors, far exceeding those even with the aid of external ferroelectrics ($I_{\text{light},\parallel} = \sim 0.3$ nA, *Nat. Commun.* 2022, 13, 3198) and amplification circuitry ($I_{\text{light},\parallel} = 7.4$ nA, *Nat. Commun.* 2022, 13, 3198). To address the potential concerns of the readership, we have modified the corresponding sentence on Page 8, Lines 219-221 as: “E.g., while $I_{\text{light},\perp}$ is depleted in the subthreshold region at $V_G = 20$ V, $I_{\text{light},\parallel}$ can still be amplified to a high level above the threshold due to the polarization-dependent ΔV_{th} , giving rise to a high DR of over 10^4 .”

Getting back to the issue of the low $I_{\text{light},\perp}$ that might be “undetectable”, we consider that this in turn proves the unique superiority of our OPT over conventional polarization-sensitive photodetectors, as high-quality polarizers also exhibit total signal extinction under orthogonally polarized light. In addition, thanks to the depleted off-state current, our OPT has a low noise equivalent power down to 0.4 fW Hz^{-0.5}. With further decreasing the noise level and the dark current, the OPT offers remarkable opportunities in the field of weak light detection and bionic visual perception (*Nat. Commun.* 2019, 10, 1294; *Nat. Electron.* 2021, 4, 522-529).

When considering practical measurements, our OPT possesses an extraordinary characteristic of gate tunability. E.g., when operating at $V_G = 10$ V, we obtained higher $I_{\text{light},\parallel}$ and $I_{\text{light},\perp}$ of $\sim 35,000$ nA and ~ 760 nA, respectively (Supplementary Fig. 24). Though at the expense of decreased polarization sensitivity, it is still sufficient for certain applications. Further improvement of the photocurrent level could be achieved by using materials with higher mobility, or reading signals through a current amplifier. On the other hand, we also offer a threshold voltage measuring mode in our device, as the threshold voltage shift is also polarization-dependent, this could help greatly simplify the signal extraction in a practical case.

Table R1 | Comparisons of $I_{\text{light},\parallel}$ and $I_{\text{light},\perp}$ that contribute to the highest polarization sensitivity in the state-of-the-art polarization-sensitive photodetectors.

Device composition	$I_{\text{light},\parallel}$ (nA)	$I_{\text{light},\perp}$ (nA)	Ref.
C8-BTBT crystal array	21,000	1.8	This work
Si NW	~ 3.6	~ 1.5	Nanoscale 2014, 6, 11232-11239
Carbon nanotubes	~ 0.5	~ 0.08	Nano Lett. 2012, 12, 5649-5653
GeSe flake	$\sim 2,600$	$\sim 1,200$	J. Am. Chem. Soc. 2017, 139, 14976-14982
TiS ₃ nanoribbon	160	40	Nanotechnology 2018, 29, 184002
Te flake	$\sim 9,000$	$\sim 1,000$	Nat. Commun. 2020, 11, 2308
Sb ₂ Se ₃ nanosheet	82	5.2	Adv. Mater. 2017, 29, 1700441
(BA) ₂ PbI ₄ NWs	~ 40	~ 10	Adv. Optical Mater. 2019, 7, 1900039

CsPbBr ₃ NW array	~360	~140	Adv. Mater. 2017, 29, 1605993
BP/WSe ₂	~18	~3	Nano Energy 2017, 37, 53-60
BP/InSe	~2.6	~0.2	Adv. Funct. Mater. 2018, 28, 1802011
P(VDF-TrFE)/BP	~0.3	~0.001	Nat. Commun. 2022, 13, 3198
Bi ₂ Se ₂ S integrated with OFET	7.4	0.02	Nat. Commun. 2021, 12, 6476

ii) the operational voltages are generally as high as several tens of volts and the response time of the devices is quite slow;

Reply: We thank the reviewer for raising this concern. The high operation voltage problem of our device can be addressed by replacing the SiO₂ layer with a high-κ dielectric layer. This could also help further prove the universality of our strategy. Accordingly, we performed a series of new experiments and fabricated the low-operation voltage OPT based on blade-coated C8-BTBT on Al₂O₃, which operates at $V_{DS} = -2$ V and a V_G range within 5 V, detailed descriptions are presented in Supplementary Fig. 36 in the revised manuscript.

Supplementary Figure 36 | a, Device structure of the low-operation voltage OPT based on blade-

coated C8-BTBT microribbons on Al₂O₃. 50 nm Al₂O₃ gate dielectric was directly grown on heavily n-doped Si through atomic layer deposition, and 1.5 nm MoO₃ layer was thermally evaporated in the electrode contact region to facilitate hole injection and improving device ideality¹⁷. **b and c**, CPOM images of the low-operation voltage OPT at different sample rotation angles. **d**, AFM characterization and **e**, the corresponding height profile of the Al₂O₃/Si boundary, showing ~50 nm thickness of the Al₂O₃ gate dielectric layer. **f**, Frequency-dependent capacitance of the Al₂O₃ gate dielectric layer at room temperature. The inset is a photograph of the n⁺⁺ Si/Al₂O₃/Ag sample with an electrode area of 0.9×0.9 cm². **g**, Transfer curves of the low-operation voltage OPT in the dark ($V_{DS} = -2$ V). **h**, Polarization-dependent transfer curves of the low-operation voltage OPT under 365 nm polarized light with an intensity of 110 μW cm⁻² ($V_{DS} = -2$ V). **i**, Dependence of ΔV_{th} on polarization angle of the low-operation voltage OPT, showing a small $\Delta V_{th,PD}$ of ~2.1 V.

In addition, we have added a corresponding description on Page 11, Lines 294-304 in the revised manuscript: “More importantly, we further demonstrated the feasibility of the anisotropic charge trapping effect on a different gate dielectric. Specifically, we fabricated a low-operation voltage OPT based on blade-coated C8-BTBT microribbons on high- κ Al₂O₃ (Supplementary Fig. 36a-g). Thanks to the considerable charge trapping capability of Al₂O₃^{49,57}, the OPT has a high DR of 2.4×10^3 with a small $\Delta V_{th,PD}$ of ~2.1 V (Supplementary Fig. 36h,i). Given the high capacitance of the Al₂O₃ layer ($C_i = 1.1 \times 10^{-7}$ F cm⁻², Supplementary Fig. 36f), the polarization-dependent difference in the trapped electron density was estimated to be $\sim 1.4 \times 10^{12}$ cm⁻² on Al₂O₃, which is comparable to that on SiO₂. Based on the aforementioned experiments, we reason that a significantly boosted DR and much lower power consumption should be envisaged through further optimization of material choice, device fabrication, and wavelength selection in a series of follow-up works.”

In terms of the slow response time issue, we have already given a detailed discussion on Page 10, Lines 281-268 and Page 13, Lines 364-370 in the main text. A coin always has two sides, and different types of devices have their respective strengths and shortcomings with specific application scenarios. Compared with fast-switching photodiodes, the tremendous amplification of polarization sensitivity in our OPT is often achieved at the cost of photoresponse speed. However, the anisotropic charge trapping effect endows the OPT with a unique anisotropic photoadaptive characteristic capable of mimicking the bionic visual perception system (such light intensity-dependent and gate voltage-dependent photoresponse behaviors under unpolarized light has been recently reported in *Nat. Electron.* 2021, 4, 522-529 and *Nat. Electron.* 2022, 5, 84-91). In addition, the persistent charge storage accumulation under weak light makes our OPT particularly sensitive in low-light environments (*Nat. Commun.* 2019, 10, 1294), offering a great promise in artificial cognition, bionic night vision, and all those polarimetric measurements that require a long integration time.

iii) though the nice picture of compact on-chip device with encapsulation is shown, it should still be carefully fixed and connected with a semiconductor parameter analyzer for outdoor measurements,

which, from my point of view, is not more advanced/convenient compared with the examples described in the manuscript as ‘requires bulky and complicated optical systems that add to the manufacturing cost’, even on the contrary, more operatively complex.

Reply: We thank the reviewer for raising this concern. As a proof-of-concept demonstration, we performed the outdoor polarization navigation measurement using our single-component OPT and showed that they could indeed realize the detection of partially polarized light in a real, complex environment. This is a significant step towards simplified on-chip polarimetry, as our central point lies in the much-simplified device structure in our proposed strategy, which does not need external optics including spatially separated polarizer and light filter as required in a conventional polarimeter, thus enabling a low-cost, facile route towards ultrasensitive polarimetry. In addition, as mentioned above, through optimization of materials and device fabrication, low-voltage devices operated at $V_{DS} = -2$ V and a V_G range within 5 V were fabricated, making possible the operation of our devices with just a few dry batteries in a series of follow-up works in the future.

Response to reviewer #2:

Reviewer's comments: The paper reports on the polarization-sensitive phototransistor by applying an anisotropic charge trapping effect, realizing a high dichroic ratio. The key point to realize it is the interface and well-aligned organic single-crystal array. Moreover, the potential application for bionic navigation is demonstrated. The work is well conducted and clearly written. I would therefore suggest the publication of the work, after addressing the following questions.

Reply: We greatly appreciate the reviewer's positive comments on our work. Our response to the reviewer's specific comments is given below:

1. Based on equation 1, ΔV_{th} is proportional to ηAP , while ΔV_{th} is also proportional to ΔN_{trap} . Does it mean all the photogenerated carriers are trapped? Is a conversion coefficient needed between the photogenerated carriers and trapped carriers?

Reply: We thank the reviewer for the careful reading of our manuscript. As illustrated in Fig. R1, equation 1 describes the nonlinear relationship between ΔV_{th} and P (*Adv. Mater.* 2013, 25, 4267-4295; *Adv. Mater.* 2018, 30, 1705542), which indicates that not all the photogenerated carriers can be trapped under illumination, because the number of trapped charge carriers will gradually saturate with increasing the light power due to the finite trap sites and the increased electron-hole recombination rate. If $\eta q \lambda P / (I_0 h c) \ll 1$ at quite low incident light power, then ΔV_{th} is to some extent proportional to ηP ; if $\eta q \lambda P / (I_0 h c) \gg 1$ under strong light, then ΔV_{th} is proportional to $\ln(\eta P)$. Since the model ideally predicts this saturation effect, we think that an additional conversion coefficient is not necessary for equation 1.

Fig. R1 | Schematic illustration of the nonlinear relationship between threshold voltage shift and the incident light power in the photovoltaic mode of a phototransistor.

For further clarification of the physical meaning of this equation, we have checked it in literature, which was originally derived by Chen et al. (*Appl. Phys. Lett.* 1981, 39, 979-981) to analyze the potential barrier lowering ($\Delta\Phi_b$) caused by minority charge carrier accumulation under illumination, where the flux of minority charge carriers leaving and moving toward the potential minimum should be balanced in the steady state:

$$\Delta N = N_0 + \Delta N'$$

where ΔN is the number of the accumulated minority charge carriers upon illumination, N_0 is the background minority charge carriers, and $\Delta N'$ is the excess photogenerated minority charge carriers. Therefore, the above expression can be further written as:

$$\frac{\Delta N}{\tau_{\text{eff}}} = \frac{I_d}{q} + \frac{\eta P}{hc/\lambda}$$

where τ_{eff} is the effective lifetime of minority charge carriers, I_d is the dark current of minority charge carriers, q is the elementary charge, η is the quantum efficiency for photogeneration, P is the light power, h is the Planck's constant, c is the speed of light in vacuum, and λ is the wavelength of the incident light. Note that:

$$\frac{1}{\tau_{\text{eff}}} = \frac{1}{\tau_r} + \frac{1}{\tau_t} + \frac{1}{\tau_{te}} + \frac{1}{\tau_D}$$

where τ_r , τ_t , τ_{te} , and τ_D are the lifetimes relating to the processes of recombination, trapping, thermionic emission, and diffusion, respectively. Therefore, τ_{eff} is actually a conversion parameter that describes the chance of the minority charge carriers being trapped; if τ_{eff} is longer, then more minority charge carriers could remain in trap sites. In the meantime, following the Boltzmann statistics, the potential energy changes linearly with the number of charge carriers changing exponentially:

$$\frac{\Delta N}{\tau_{\text{eff}}} = C \exp\left(\frac{q\Delta\Phi_b}{nkT}\right)$$

where C is a proportionality constant, n is a constant that describes the saturation effect, k is the Boltzmann constant, and T is the temperature. Therefore, the combination of the above expressions leads to:

$$\Delta\Phi_b = \frac{nkT}{q} \ln\left(1 + \eta \frac{q\lambda P}{I_d hc}\right)$$

With ΔV_{th} assumed to be proportional to $\Delta\Phi_b$ (*IEEE Electron Device Lett.* 1998, 19, 472-474; *IEEE Trans. Electron Devices* 1999, 46, 2271-2277; *Adv. Mater.* 2018, 30, 1705542), the light power-dependent ΔV_{th} can be expressed as:

$$\Delta V_{\text{th}} = \frac{NkT}{q} \ln\left(1 + \eta \frac{q\lambda P}{I_d hc}\right)$$

where N is an empirical constant.

2. In page 9, can the authors explain the reason that the trap site filling time is relatively long (>5s) and different from the light intensity?

Reply: The relatively long trap site filling time is commonly observed in phototransistors with abundant trap sites (*Adv. Opt. Mater.* 2016, 4, 264-270; *Adv. Funct. Mater.* 2019, 29, 1905657; *Nat. Commun.* 2019, 10, 12). Especially, for organic semiconductors, the weakly bonded molecular interactions make them particularly sensitive to the surrounding environment (e.g., water and oxygen) and the interfacial active groups, such as $-\text{OH}$, $-\text{NH}_2$, and $-\text{COOH}$, ensuring sufficient trap sites that require a long time to be fully occupied. Besides, trap sites are commonly energetically distributed,

with deeper traps being occupied preferentially at first, followed by the gradual filling of shallower traps (*Nat. Commun.* 2018, 9, 4546). Under light illumination, trapping and detrapping processes could take place at the same time. Consequently, the acumination time has to be sufficiently long to realize the complete saturation of trap sites.

One typical characteristic of photodetectors with charge trapping effect is the light intensity-dependent response time, because the degree of trap occupancy is determined by the light intensity (*Nat. Nanotechnol.* 2012, 7, 798-802), or in other words, the number of photogenerated charge carriers that occupy trap sites. As illustrated in Fig. R2a,b, compared with stronger light that leads to faster saturation of trap site filling, weaker light has smaller photon flux per unit of time, thus generating fewer excitons within the photoactive layer and resulting in smaller chance of electrons occupying the trap sites. Therefore, it takes a longer time to fully occupy the trap sites under weaker light.

Intriguingly, a few recent works have focused on this relatively “slow” response behavior of phototransistors towards bionic applications (*Nat. Electron.* 2021, 4, 522-529, *Nat. Electron.* 2022, 5, 84-91). With the aid of the gate controllability, the light intensity-dependent photoexcitation and inhibition behaviors in phototransistors can be readily tuned to mimic the visual perception process of human eyes, which shares similar self-adaptation speeds from several seconds to tens of seconds under light stimulus. Furthermore, with further increasing the exposure time, quite weak light signals could be detected due to the greatly strengthened gating effect caused by the gradual filling of trap sites (*Nat. Commun.* 2019, 10, 1294). Therefore, compared with conventional fast-switching photodiodes, the tunable photoresponse characteristics make our OPTs particularly useful in novel artificial cognition applications, especially for acquiring weak signals under low-light environments.

Fig. R2 | Schematic illustrations of the charge trapping process under **a**, weaker light and **b**, stronger light.

3. Also, one can detrapp the electrons by applying an erasing gate voltage to achieve a fast recovery. How to increase the response speed or the trap speed?

Reply: We thank the reviewer for this valuable comment. Since a negative erasing V_G could help realize a fast recovery, one possible route to boost the trapping process is using a more positive V_G . Accordingly, we have added a more detailed description in Supplementary Fig. 17 in the revised manuscript. By applying $V_G = -25$ V under illumination, I_{DS} exhibits persistent increase due to the

competition between light-triggered electron trapping and V_G -facilitated electron detrapping. In contrast, applying $V_G = 25$ V under illumination would lead to positive charge polarity on the top of the dielectric layer, which makes it easier to induce more electrons in trap sites, thus increasing the response speed and realizing faster saturation of I_{DS} (Supplementary Fig. 17a). To analyze the V_G -dependent electron trap rate, we measured the transfer curves in the dark after the illumination of UV light for 10 s at different V_G (Supplementary Fig. 17b). While the initial dark state maintained stable, applying a more positive V_G under illumination could lead to a much more positively drifted transfer curve. Based on the extracted ΔV_{th} , the trap rates were calculated to be dependent on the applied V_G (Supplementary Fig. 17c).

In the meantime, we have modified the corresponding descriptions on Page 7, Lines 179-182 in the revised manuscript: “We note that the filling of trap sites is associated with the rise of I_{DS} when switching UV light on⁴⁸, which shows a unique photoadaptation characteristic that is tunable by both light intensity and the applied V_G (Supplementary Fig. 16f and Supplementary Fig. 17).”

Supplementary Figure 17 | a, Time-related evolution of I_{DS} upon turning on/off the unpolarized UV light with an intensity of $135 \mu\text{W cm}^{-2}$ at different V_G ($V_{DS} = -40$ V). **b**, Transfer characteristics of the OPT measured in the dark after the illumination of $135 \mu\text{W cm}^{-2}$ UV light for 10 s at different V_G ($V_{DS} = -40$ V). Applying a more positive V_G under illumination helps facilitate electron trapping, leading to a larger threshold voltage shift. **c**, V_G -dependent average electron trap rates under 10 s of UV illumination. The average trap rates were estimated according to $\Delta N_{\text{trap}}/\Delta t = (C_i \Delta V_{\text{th}})/(q \Delta t)^{16}$, where Δt is the illumination time, and ΔV_{th} is the corresponding threshold voltage shift after 10 s of UV illumination in (b).

We further note that, besides the rational selection of an appropriate V_G , further pathways to the optimization of photoresponse speed should include the improvements of crystal quality, device geometry, and fabrication process, in order to suppress undesired charge carrier recombination and increase the transit time of majority charge carriers. On the other hand, performing surface trap functionalization on the dielectric layer in a more uniform and controllable manner could help facilitate charge trapping. *E.g.*, colloidal quantum dots or nanoparticles are good candidates to replace the hydroxyl groups on SiO_2 , as their size, ligand, and oxidation could be tuned and engineered more easily to control trap site properties (*Nature* 2006, 442, 180-183; *Nat. Nanotechnol.* 2012, 7, 798-802).

Response to reviewer #3:

Reviewer's comments: This paper uses the defect structure of the silicon dioxide layer on the silicon substrate to construct a facile charge trapping effect, which can be combined with the single crystal active layer to achieve the anisotropy charge trapping, which is very innovative and interesting, but there are still some problems in the mechanism analysis and experimental methods, and it is suggested to be published after major revision.

Reply: We greatly appreciate the reviewer's positive comments on our work. Our response to the reviewer's specific comments is given below:

1, please confirm whether formula 1 is written correctly. More often, we use this formula to describe photocurrent (I_{ds}), which will be different with by one transconductance factor from the current expression.

Reply: We thank the reviewer for the careful reading of our manuscript. We would like to confirm that formula 1 is correctly written in our manuscript. After checking its original form in literature, we note that both expressions of ΔV_{th} and ΔI_{DS} are widely adopted to describe the non-linear photovoltaic mode of phototransistors under light illumination (*Polym. Int.* 2012, 61, 374-389; *Adv. Mater.* 2013, 25, 4267-4295; *Adv. Mater.* 2018, 30, 1705542). In our manuscript, since I_{DS} and transconductance (g_m) vary with the applied gate voltage, it should be much more explicit to use the expression of ΔV_{th} to quantitatively analyze the polarization-dependent transfer curve shift.

To gain more insight into the physical meaning of this formula, we have provided more details as follows. The model was first developed by Chen et al. (*Appl. Phys. Lett.* 1981, 39, 979-981) to describe the potential barrier lowering ($\Delta\Phi_b$) caused by the accumulation of minority charge carriers under illumination. The balance of the flux of minority charge carriers in the potential minimum requires that the number of the accumulated minority charge carriers (ΔN) should be equal to the sum of background minority charge carriers and the excess photogenerated minority charge carriers under illumination:

$$\frac{\Delta N}{\tau_{eff}} = \frac{I_d}{q} + \frac{\eta P}{hc/\lambda}$$

where τ_{eff} is the effective lifetime of minority charge carriers, I_d is the dark current of minority charge carriers, q is the elementary charge, η is the quantum efficiency for photogeneration, P is the light power, h is the Planck's constant, c is the speed of light in vacuum, and λ is the wavelength of the incident light. On the other hand, the number of the accumulated minority charge carriers is related to barrier lowering following the Boltzmann statistics, which can be expressed as:

$$\frac{\Delta N}{\tau_{eff}} = C \exp\left(\frac{q\Delta\Phi_b}{nkT}\right)$$

where C is a proportionality constant, n is a constant relating to the saturation effect, k is the Boltzmann constant, and T is the temperature. Therefore, the combination of the above two expressions leads to:

$$\Delta\Phi_b = \frac{nkT}{q} \ln\left(1 + \eta \frac{q\lambda P}{I_d h c}\right)$$

Further developments of this model were conducted by Takanashi et al. (*IEEE Electron Device Lett.* 1998, 19, 472-474; *IEEE Trans. Electron Devices* 1999, 46, 2271-2277) to describe the photovoltaic mode of InAlAs/InGaAs phototransistors, where the threshold voltage shift (ΔV_{th}) was assumed to be proportional to $\Delta\Phi_b$, which leads to the following equation:

$$\Delta V_{th} = \frac{NkT}{q} \ln\left(1 + \eta \frac{q\lambda P}{I_d h c}\right)$$

where N is an empirical constant. Consequently, on the basis of the ΔV_{th} - P relationship, the change of photocurrent (ΔI_{DS}) was further deduced to be:

$$I_{DS} = g_m \Delta V_{th} = \frac{AkT}{q} \ln\left(1 + \eta \frac{q\lambda P}{I_d h c}\right)$$

where g_m is the transconductance of the phototransistor and A is a proportionality parameter.

2. The current formulae 1,2 are mainly concerned with the anisotropy of crystals. Why can we see the difference in anisotropy charge trapping from these formulas? Please explain further, for example, which parameter it belongs to.

Reply: We thank the reviewer for raising this concern. The shift of threshold voltage (ΔV_{th}) under light illumination is considered a parameter directly showing the consequence of charge trapping in phototransistors, with the amount of trapped charge carrier density estimated by (*Nat. Commun.* 2019, 10, 756, *Nat. Commun.* 2019, 10, 12, *Nat. Electron.* 2022, 5, 84-91):

$$\Delta N_{trap} = \Delta V_{th} C_i / q$$

where C_i is the unit area capacitance of the gate dielectric layer. Formula 1 describes the light power-dependent ΔV_{th} of phototransistors under illumination. If we take the anisotropic light absorption of organic crystals into account, for orthogonally polarized light with \parallel and \perp states representing the maximum and minimum light absorption directions, respectively, the corresponding threshold voltage shifts can be expressed as:

$$\Delta V_{th,\parallel} = \frac{NkT}{q} \ln\left(1 + \frac{q\lambda}{I_d h c} a_{in} P_{eff}\right)$$

$$\Delta V_{th,\perp} = \frac{NkT}{q} \ln\left(1 + \frac{q\lambda}{I_d h c} P_{eff}\right)$$

where a_{in} is the anisotropic ratio of light absorption and P_{eff} is the effective light power. Therefore, the difference in threshold voltage shift between \parallel and \perp states can be expressed by Formula 2 in the main text:

$$\Delta V_{th,\parallel} - \Delta V_{th,\perp} = \frac{NkT}{q} \left[\ln\left(1 + \frac{q\lambda}{I_d h c} a_{in} P_{eff}\right) - \ln\left(1 + \frac{q\lambda}{I_d h c} P_{eff}\right) \right]$$

The polarization-dependent threshold voltage shift ($\Delta V_{th,PD} = \Delta V_{th,\parallel} - \Delta V_{th,\perp}$) was thus used to quantitatively analyze the anisotropic charge trapping under orthogonally polarized light. To make our

explanations clearer, we have modified the corresponding sentences on Page 5, Lines 124-129 in the revised manuscript: “For instance, a photoactive crystal with $a_{in} = 5.6$ (which is the value of the organic crystal we used for fabricating the OPT in following discussions) may cause a $\Delta V_{th,PD}$ up to 34.2 V (Supplementary Fig. 3). Accordingly, a large difference in the trapped electron density of $\sim 3.8 \times 10^{12} \text{ cm}^{-2}$ was estimated on a 200 nm-thick SiO_2 dielectric layer ($\Delta N_{\text{trap}} = \Delta V_{\text{th,PD}} C_i / q^{41}$, where C_i represents the unit-area capacitance of the dielectric layer), indicating the strong anisotropy of charge trapping under orthogonally polarized light.”

3. In fact, the change of threshold voltage is relatively small, and the corresponding photocurrent (I_{ds}) change is very large, does this result indicate that the transconductance effect of anisotropy is also different.

Reply: We thank the reviewer for raising this concern. We agree with the reviewer that the transconductance effect should also be taken into consideration as it reflects the current amplification capability of the OPT. Following the reviewer’s suggestion, we calculated the transconductance of the OPT according to $g_m = d\sqrt{I_{DS}}/dV_G$ in the saturation regime of an organic field-effect transistor (*Adv. Funct. Mater.* 2019, 29, 1906653), details have been added in Supplementary Fig. 25. Intriguingly, we found a polarization-dependent trend of the peak g_m under polarized light, indicating that aside from the drift of transfer curves, the difference in the increasing levels of I_{DS} further helps to boost the polarization sensitivity. The increase of peak g_m with light intensity has been observed by a few works (*Nature* 2018, 561, 516-521, *Appl. Phys. Lett.* 2014, 105, 232105), though the mechanism needs further exploration, it could be related to the improved channel conductance and the enhanced gating effect under stronger light illumination. Furthermore, we also evaluated g_m -related starting voltages (V_{st} , the gate voltage where g_m starts to increase) and the saturation voltages (V_{sa} , the gate voltage where g_m reaches its peak). These parameters are independent of the conventional V_{th} extraction method, thus providing further evidence for anisotropic charge trapping. Compared with the irregular fluctuations in the dark, V_{st} and V_{sa} under polarized light also sinusoidally vary with polarization angles similar to V_{th} .

Accordingly, we added more detailed descriptions and discussions on Pages 8-9, Lines 221-232:

“In the meantime, we derived the V_G -dependent transconductance ($g_m = d\sqrt{I_{DS}}/dV_G$ in the saturation regime of an OFET) to analyze the gate-controlled current amplification (Supplementary Fig. 25a,b). The increase of g_m with V_G becoming more negative is due to the reduced hole injection barrier when the OPT is switched on. On the other hand, with the existence of trap sites, g_m becomes saturated and decreases at a higher negative V_G . Intriguingly, while the initial dark-state g_m has no anisotropy, the gate voltages where g_m starts to increase (V_{st}) and reaches its peak (V_{sa}) exhibit a polarization-dependent trend similar to that of ΔV_{th} under polarized light (Supplementary Fig. 25c,d). Furthermore, the peak g_m is also polarization-dependent with $I_{\text{light}, \parallel}$ rising more steeply than $I_{\text{light}, \perp}$ (Supplementary

Fig. 25e), which is possibly caused by the increased charge carrier concentration in the conductive channel and the enhanced gating effect under stronger light absorption. Accordingly, we reason that this unique anisotropic amplification characteristic further endows the OPT with a greatly boosted polarization sensitivity.”

Supplementary Figure 25 | a and b, Gate voltage-dependent g_m of the OPT before and under polarized light illumination, respectively. **c-e**, Polarization-dependent V_{st} , V_{sa} , and peak g_m , in the dark (grey curves) and under polarized light illumination (red curves), respectively.

4. The completely different charge trapping characteristics of materials on SiO_2 and CYTOP are good evidence for the effect of interface defects. Is there further quantitative results, such as the difference of subthreshold swing amplitude and channel defect density between the two anisotropy charge trapping?

Reply: We thank the reviewer for the positive comment. Following the reviewer’s suggestion, we further conducted a number of experiments to quantitatively show the influence of interfacial defect density on anisotropic charge trapping. Because CYTOP is a superlyophobic material unsuitable for ordered crystal growth, we utilized two other insulating materials, divinyltetramethylsiloxane-bis(benzocyclobutene) (BCB) and polystyrene (PS) to passivate the trap sites on SiO_2 in a solution grown process. The resulting C8-BTBT crystals are shown in Fig. R3a,b and Fig. R3d,e, respectively. The uniform colors under cross-polarized optical microscopy (CPOM) indicate that well-oriented C8-BTBT crystals were prepared for device fabrication. However, due to the passivated C8-BTBT/ SiO_2 interface by BCB and PS, both devices exhibit negligible transfer curve shift under polarized light, resulting in much lower polarization sensitivity (Fig. R3c,f).

We next calculated the defect density of these devices based on the following equation (*Science* 2019, 363, 719-723; *Adv. Electron. Mater.* 2021, 7, 2100105):

$$N_{\text{defect}} = \frac{C_i}{q} \left(\frac{qSS}{kT \ln 10} - 1 \right)$$

where SS is the subthreshold swing defined by $dV_G/d(\log I_{DS})$, and C_i values of BCB/SiO₂ and PS/SiO₂ were extracted from our previous works (*Adv. Funct. Mater.* 2021, 31, 2105459, *Adv. Funct. Mater.* 2021, 31, 2100202). Table R2 lists the dark-state subthreshold swing (SS_{dark}) of the devices based on C8-BTBT crystal array/SiO₂, C8-BTBT:PS crystal/SiO₂, and C8-BTBT crystal/BCB/SiO₂, which clearly reveals that a much smaller SS_{dark} could be attained through passivation of the interfacial defects by PS or BCB. As a result, N_{defect} was lowered by over one order of magnitude compared with that on bare SiO₂. In the meantime, the polarization-dependent threshold voltage shifts ($\Delta V_{\text{th,PD}}$) of the OPTs based on C8-BTBT:PS crystal/SiO₂ and C8-BTBT crystal/BCB/SiO₂ are only ~4 V and ~0.7 V, respectively, which are far smaller than that of C8-BTBT crystal array/SiO₂-based OPT (~25 V, see Fig. R4a and Table R2 for more details). Consequently, these passivated devices have lower density of trapped charge carriers under polarized light ($\Delta N_{\text{trap,PD}} = \Delta V_{\text{th,PD}} C_i / q$, Table R2).

Intriguingly, while the passivated devices only show slight deviation of subthreshold swing under polarized light (SS_{light}) due to fewer trapped charge carriers, the SS_{light} of the C8-BTBT crystal array/SiO₂-based OPT exhibited a polarization-dependent trend under polarized light (Fig. R4b). The lowering of SS_{light} with increasing the number of trapped charge carriers could be a consequence of shielding interfacial defects (*IEEE Trans. Electron Devices* 2020, 67, 3645-3649) and the increment of effective gate control (*2021 IEEE 14th International Conference on ASIC*, DOI: 10.1109/asicon52560.2021.9620324). However, it should be further noted that several factors, such as the measurement accuracy and the level of off-state current could also affect SS. *E.g.*, a significant increase in the channel conductivity under light illumination would lead to an increased SS_{light} (*MRS Proceedings* 2003, 771, 1017). Therefore, SS_{light} is rarely used as a reference standard in literature, more often we compare the change of threshold voltage shift to evaluate the number of trapped charge carriers under light illumination.

Fig. R3 | **a and b**, CPOM images of the C8-BTBT crystal on BCB at sample rotation angles of 45° and 0° , respectively. **c**, Polarization-dependent transfer curves of the OPT based on C8-BTBT crystal/BCB/SiO₂. **d and e**, CPOM images of the C8-BTBT:PS crystal on SiO₂ at sample rotation angles of 45° and 0° , respectively. **f**, Polarization-dependent transfer curves of the OPT based on C8-BTBT:PS crystal/SiO₂.

Table R2 | Comparisons of the anisotropic charge trapping behaviors of C8-BTBT crystal-based OPTs with different device structures. Values of SS_{dark} and $\Delta V_{\text{th,PD}}$ of the OPT based on C8-BTBT crystal array/SiO₂ were extracted from Supplementary Fig. 28 and Supplementary Fig. 26b, respectively.

Device structure	C_i (F cm ⁻²)	SS_{dark} (V dec ⁻¹)	N_{defect} (cm ⁻²)	$\Delta V_{\text{th,PD}}$ (V)	$\Delta N_{\text{trap,PD}}$ (cm ⁻²)
C8-BTBT crystal array/SiO ₂	1.8×10^{-8}	3.9	7.3×10^{12}	~ 25.1	2.8×10^{12}
C8-BTBT:PS crystal/SiO ₂	7.4×10^{-9}	0.8	5.8×10^{11}	~ 4.0	1.9×10^{11}
C8-BTBT crystal/BCB/SiO ₂	5.0×10^{-9}	0.9	4.4×10^{11}	~ 0.7	2.2×10^{10}

Fig. R4 | Polarization-dependent **a**, threshold voltage shift and **b**, subthreshold swing of C8-BTBT crystal-based OPTs with different device structures.

5. Why the charge capture using SiO₂ can only be seen under light? SiO₂ in dark state has defects. Whether the SiO₂ used in this paper has been processed or prepared by a special method.

Reply: We thank the reviewer for the careful reading of our manuscript. The SiO₂/Si wafers used in this work were purchased from Seiren KST Corp. and dealt with sequential cleaning and ozone treatment before usage. The –OH groups that function as trap sites could naturally exist on SiO₂ without further special treatment. We note that although abundant trap sites exist on SiO₂, no extra electrons can be captured in the dark given the p-type nature of C8-BTBT without external electron injection. This unobvious electron injection phenomenon could be related to the relatively large bandgap of C8-BTBT and the high-quality C8-BTBT/Au contact region, which prohibits the injection of excess electrons from electrodes (discussions on the bias stress stability of the device in the dark have been shown in Supplementary Fig. 29). The minority charge carrier injection phenomenon was systematically studied in our previous work (*Adv. Funct. Mater.* 2021, 31, 2100202), where smaller-bandgap dif-TES ADT (~2.1 eV) exhibited obvious positive drift of transfer curves in the dark due to electron injection. When decorating dif-TES ADT with larger bandgap C10-BTBT (~3.4 eV) in the electrode contact region, electron injection was suppressed with greatly improved bias stress stability in the dark. In this manuscript, the large bandgap of C8-BTBT and the well-prepared C8-BTBT/Au contact region (Au deposition rate = ~0.1 Å/s) ensured that no extra electrons could be injected in the dark. On the other hand, illumination by UV light would generate a large number of electrons within C8-BTBT, which could easily be captured by the trap sites on SiO₂, leading to obvious drift of transfer curves.

To perform more detailed investigations on the minority charge carrier injection issue of C8-BTBT, we further presented a series of experiments showing the influence of the C8-BTBT/Au contact region. As illustrated in Fig. R5a, a carefully tuned Au deposition rate (*i.e.*, ~0.1 Å/s) would result in high-quality C8-BTBT/Au contact region with few thermal damages, in this case, electrons could be hardly injected due to the large bandgap and high crystal quality of C8-BTBT, ensuring the bias stress stability of the transfer curve in the dark (Fig. R5b). However, at higher Au deposition rate (~0.5 Å/s), C8-BTBT with low-temperature liquid crystal property would easily be melted and damaged (Fig. R6a,b), which would introduce defects that facilitate electron injection and trapping (Fig. R5c), resulting in transfer curve drift in the dark (Fig. R5d). Accordingly, we have added an description in the caption of Supplementary Fig. 14: “The negligible hysteresis in the dark is attributed to the high crystal quality and large bandgap of C8-BTBT, which prohibit undesired electron injection in the dark¹⁵.”

Fig. R5 | **a**, Schematic illustration of the band alignment in the high-quality C8-BTBT/Au contact region, indicating no electron injection under positive gate bias voltage due to the large bandgap of C8-BTBT. **b**, Transfer curves of the C8-BTBT crystal array-based phototransistor measured with different starting gate voltages ($V_{DS} = -40$ V), Au deposition rate ~ 0.1 Å/s. **c**, Schematic illustration of the band alignment in the thermally damaged C8-BTBT/Au contact region, the introduction of defects will easily lead to electron injection and trapping under positive gate bias voltage. **d**, Transfer curves of the C8-BTBT crystal array-based phototransistor measured with different starting gate voltages ($V_{DS} = -40$ V), Au deposition rate = ~ 0.5 Å/s. The HOMO level of C8-BTBT was extracted from previous reports, with the LUMO level determined by the difference between the HOMO level and the optical bandgap of C8-BTBT (*J. Am. Chem. Soc.* 2007, 129, 15732-15733; *Adv. Funct. Mater.* 2015, 25, 5669-5676).

Fig. R6 | **a and b**, Typical SEM images of the C8-BTBT crystal array/Au contact region with Au deposition rate = ~ 0.5 Å/s, showing severe thermal damages.

6. Slow response is a big problem for this kind of device. The paper only gives rise time, what is the

descent time of this kind of device?

Reply: We thank the reviewer for raising this concern. We note that the tremendous photocurrent amplification capability of the OPT is indeed achieved at the cost of photoresponse speed. However, on the other hand, the unique illumination time-dependent polarized photoresponse endows the OPT an extraordinary photoadaptive characteristic that is especially useful for bionic visual perception (*Nat. Electron.* 2021, 4, 522-529; *Nat. Electron.* 2022, 5, 84-91). Particularly, the persistent amplification of polarized signals under weak light promises major advances in low-light cognition (*Nat. Commun.* 2019, 10, 1294). Based on the deep understanding of our OPT, we have provided detailed discussions on Page 10, Lines 281-286 and Page 13, Lines 364-374 in the main text. More importantly, to prove the robustness of our OPT under weak, partially polarized light, we have successfully demonstrated a celestial compass in a real, complex environment, which could not be realized by conventional photodiodes or photoconductors.

In addition, following the reviewer's comment, we have further analyzed the decay of I_{DS} after turning off polarized light in detail. As depicted in Fig. 3d in the main text, due to the gradual detrapping process, the phototransistor maintained high I_{DS} that would last for hundreds of seconds. This phenomenon is commonly observed in photoconductors or phototransistors with abundant trap sites, namely the persistent photoconductivity effect (PPC, *Nat. Nanotechnol.* 2013, 8, 826-830; *Adv. Funct. Mater.* 2019, 29, 1905657). Accordingly, we utilized a bi-exponential equation to quantitatively analyze the descent times of polarization-dependent I_{DS} (*Adv. Mater.* 2014, 26, 1541-1550; *Adv. Electron. Mater.* 2015, 1, 1500119; *Adv. Funct. Mater.* 2019, 29, 1905657), the corresponding results have been added in Supplementary Fig. 31.

Supplementary Figure 31 | a-g, Enlarged polarization-dependent decay curves of Fig. 3d, showing the persistent photoconductivity upon turning off $110 \mu\text{W cm}^{-2}$ UV light with polarization angles of 0° , 15° , 30° , 45° , 60° , 75° , and 90° , respectively. The decay curves were fitted by $I_{DS} = I_0 + A_1 \exp(-t/\tau_1) + A_2 \exp(-t/\tau_2)$ with fitting curves plotted in dashed lines, where I_0 is the drain-source current before illumination is removed, A_1 and A_2 are positive fitting constants, and τ_1 and τ_2 are fast and slow decay time constants, respectively.

In the meantime, we have reorganized Fig. 3 and add a corresponding description on Page 10, Lines 261-269 in the revised manuscript: “We also notice that the OPT maintained polarization-dependent persistent photoconductivity after turning off light (Fig. 3d). The decay of I_{DS} can be described by a bi-exponential equation with time constants τ_1 and τ_2 , respectively, representing the time for fast electron-hole recombination and slow electron detrapping⁵² (Fig. 3f and Supplementary Fig. 31). The relatively shorter electron detrapping time at 0° (*i.e.*, equivalently at a higher light intensity) is related to the filling of energetically distributed trap sites. Because longer-lived deeper traps will be preferentially occupied upon illumination, they account for a slower detrapping rate under weaker light; when these deeper traps have already been occupied under strong light, filling of more shallower traps will take place, leading to a faster detrapping rate^{48,51}.”

Fig. 3 | Polarization-sensitive photodetection of the C8-BTBT crystal array. **d**, Time-related evolution of I_{DS} upon turning on/off the polarized light ($V_G = 25$ V, $V_{DS} = -40$ V). **e**, Dependence of I_{sat} and t_r on polarization angle. t_r is defined as the time required for I_{light} to increase from 10% I_{sat} to 90% I_{sat} . **f**, Dependence of decay time constants τ_1 and τ_2 on polarization angle.

In order to eliminate the influence of the PPC effect, a negative V_G can be applied to effectively boost the detrapping process (details are shown in Fig. 3g and Supplementary Fig. 32). We further provided an enlarged plot in Fig. R7 showing this transient decay process with the aid of an erasing V_G . Applying a large negative V_G (*i.e.*, -100 V in this case) upon light off helps sweep the electrons out of the trap sites, which results in the complete recovery of I_{DS} to its initial level. The estimated decay time for I_{DS} (the time required for I_{DS} to decay from 90% I_{max} to 10% I_{max}) is < 1 s, which is several hundreds of times faster than the decay in the PPC process. Accordingly, we have rephrased the corresponding sentence on Page 10, Lines 269-275 in the revised manuscript: “Fig. 3g shows the transient on/off switching behaviors of the OPT under orthogonally polarized light. Notably, the OPT not only shows an enormously high DR of over 10^4 after ~ 5 s of UV illumination, but also can be readily switched off to its initial dark current level within 1 s by applying an erasing V_G of -100 V upon turning off light (Supplementary Fig. 32), and it retains stable and repeatable polarized photoresponse within 10 on/off switching circles, showing good durability and excellent fatigue resistance.”

Fig. R7 | Enlarged plot of the transient on/off switching at 0° in Fig. 3g in the revised manuscript.

REVIEWERS' COMMENTS

Reviewer #1 (Remarks to the Author):

A great effort has been made by the authors to improve the manuscript. The response to my concerns, as well as to those from other reviewers, is very detailed and persuasive. I do not have further comments now and would like to suggest its acceptance for publication on Nat. Commun.

Reviewer #2 (Remarks to the Author):

authors properly reply all my question and comment, I recommend the paper is accepted for publication.

Reviewer #3 (Remarks to the Author):

This paper uses the defect structure of the silicon dioxide layer on the silicon substrate to construct a facile charge trapping effect, which can be combined with the single crystal active layer to achieve the anisotropy charge trapping, which is very innovative and interesting. The author has made experimental additions and modifications as required, answered relevant questions, and agreed to publish this article considering the above factors.

Responses to the reviewers' comments (Manuscript number NCOMMS-22-17386A)

Response to reviewer #1:

Reviewer's comments:

A great effort has been made by the authors to improve the manuscript. The response to my concerns, as well as to those from other reviewers, is very detailed and persuasive. I do not have further comments now and would like to suggest its acceptance for publication on Nat. Commun.

Reply: We greatly appreciate the reviewer's positive comments on our work.

Response to reviewer #2:

Reviewer's comments:

Authors properly reply all my question and comment, I recommend the paper is accepted for publication.

Reply: We greatly appreciate the reviewer's positive comments on our work.

Response to reviewer #3:

Reviewer's comments:

This paper uses the defect structure of the silicon dioxide layer on the silicon substrate to construct a facile charge trapping effect, which can be combined with the single crystal active layer to achieve the anisotropy charge trapping, which is very innovative and interesting. The author has made experimental additions and modifications as required, answered relevant questions, and agreed to publish this article considering the above factors.

Reply: We greatly appreciate the reviewer's positive comments on our work.